# ADAPTIVE TD-LAMBDA FOR COOPERATIVE MULTI-AGENT REINFORCEMENT LEARNING

## ABSTRACT

Recent advancements in multi-agent reinforcement learning (MARL) have prominently leveraged Temporal Difference Lambda, TD($\lambda$), as a catalyst for expediting the temporal difference learning process in value functions. TD($\lambda$) in value-based MARL algorithms or the Temporal Difference critic learning in Actor-Critic-based (AC-based) algorithms synergistically integrate elements from Monte-Carlo simulation and Q function bootstrapping via dynamic programming, which effectively addresses the inherent bias-variance trade-off in value estimation. Based on that, some recent works link the adaptive $\lambda$ value to the policy distribution in the single-agent reinforcement learning area. However, because of the large joint action space, the large observation space, and the limited transition data in Multi-agent Reinforcement Learning, the computation of policy distribution is infeasible to be calculated statistically. To solve the policy distribution calculation problem in MARL settings, we employ a parametric likelihood-free density ratio estimator with two replay buffers instead of calculating statistically. The two replay buffers of different sizes store the historical trajectories that represent the data distribution of the past and current policies correspondingly. Based on the estimator, we assign Adaptive TD($\lambda$), **ATD($\lambda$)**, values to state-action pairs based on their likelihood under the stationary distribution of the current policy. We apply the proposed method on two competitive baseline methods, QMIX for value-based algorithms, and MAPPO for AC-based algorithms, over SMAC benchmarks and Gfootball academy scenarios, and demonstrate consistently competitive or superior performance compared to other baseline approaches with static $\lambda$ values.

## 1 INTRODUCTION

Recent advances in Multi-agent reinforcement learning (MARL) have led to significant progress in a wide range of applications such as autonomous vehicle teams (Cao et al., 2012) and sensor networks (Zhang & Lesser, 2011). Within the MARL landscape, various value-based approaches target enhancements in either value decomposition (Sunehag et al., 2017; Rashid et al., 2018; Wang et al., 2020a) or cooperative exploration (Mahajan et al., 2019; Yang et al., 2020; Wang et al., 2020b). These prevalent methodologies involve the utilization of temporal difference (TD) updates for training the Q value function. Additionally, actor-critic methodologies, including (Foerster et al., 2018; Yu et al., 2022; Wang et al., 2023), have also exhibited outstanding performance across challenging tasks, such as StarCraft II (Samvelyan et al., 2019) and Google Football Research (Kurach et al., 2020) and these algorithms also leverage TD updates on the training of the critic network.

Nevertheless, TD learning confronts the challenge of over-estimation bias stemming from function approximation (Cicek et al., 2021), and Monte-Carlo methods introduce lower bias but exhibit larger variances (Sutton & Barto, 2018). The large bias or the large variance may make the credit assignment process unstable in the training process. Therefore, the fundamental trade-off in MARL also lies in the definition of the update target: should one estimate Monte-Carlo returns or bootstrap from an existing Q-function (Seijen & Sutton, 2014)? To flexibly navigate this trade-off between bias and variance in value estimation, the incorporation of TD($\lambda$) becomes crucial.

To demonstrate the significant impact of different $\lambda$ values in the TD($\lambda$) method on final performance, we conducted a toy experiment within a multi-agent lava-path scenario. Illustrated in Figure 1, the learning curve of the adaptive TD($\lambda$) values surpasses those of fixed lambda values. This observation

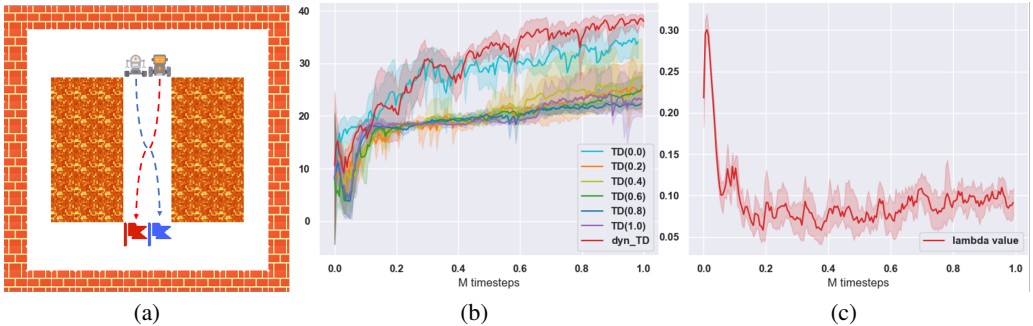

Figure 1: (a): Two agents are asked to reach their opposite goals within 60 steps without collision. Agents can choose from moving in four directions and a 'staying' action. Agents receive 40 marks when both of them reach their goals and -10 marks when they step into the lava. Otherwise, agents receive the marks of their distance to their goals subtracted by 40 after the time step limit. (b): The performance curves of different commonly-used preset TD($\lambda$) values with adaptive $\lambda$ values. The x-axis is the training time steps (e6) and the y-axis is the final performance. (c) The average adaptive lambda values during the training process.

underscores the profound influence of well-chosen $\lambda$ values in enhancing training performance while underscoring the potential detriment incurred by inappropriate $\lambda$ values. This complexity between $\lambda$ values and performances makes the choice of $\lambda$ values as hyperparameters a challenging task.

Meanwhile, proper $\lambda$ values vary intuitively based on the different training MARL process. For example, higher $\lambda$ values (almost MC) allows the target to better reflect real long-term returns, which accelerate the fitting and speeding up the stabilization of the joint value at early training stage. In contrast the policy stabilizes and replay samples from older policies become biased relative to the current policy from the middle to later training process. In such a way, smaller $\lambda$ values reduce reliance on outdated MC returns and instead trust the increasingly accurate critic. Therefore, the proper $\lambda$ value should vary according to the training process.

Based on that, we introduce the **ATD($\lambda$)**, a novel approach for determining the $\lambda$ value based on sampled transitions during training. Inspired by recent studies in density ratio calculation (Sinha et al., 2022) and off-policy policy evaluation (Grover et al., 2019), we employ a likelihood-free parametric network to simulate the density ratio for each state-action pair of a batch of sampled trajectories, and scale this ratio to serve as the adaptive $\lambda$ values. This approach utilizes a large replay buffer to store off-policy trajectories for sampling and a much smaller replay buffer to store on-policy trajectories and estimates the degree of on-policiness adherence for sampled off-policy trajectory data by calculating the $f$-divergences between the two replay buffers by the parametric network.

The main contributions of this work are: **1)** We propose an MARL-specific formulation of adaptive $\lambda$ calculation method for each transition by based on parametric likelihood-free off-policy estimation. **2)** We propose a training mechanism using two replay buffers to approximate density ratios between on- and off-policy distributions without explicit policy modeling and adapt that to existing value-based approaches and value-based critic AC algorithms with minor changes to existing MARL code-bases. **3)** We describe the feasibility of our framework from theoretical perspectives and validate our methods empirically by extensive experiments on SMAC benchmarks and Gfootball academy tasks. Experimental results indicate that existing MARL methods equipped with ATD can compete with or outperform original MARL methods in terms of the winning rates or accumulated rewards.

## 2  RELATED WORK

**Multi-agent Reinforcement Learning:**   In multi-agent value-based algorithms, the centralized value function, usually a joint Q-function, is decomposed into local utility functions. Many methods have been proposed to meet the Individual-Global-Maximum (IGM) (Son et al., 2019) assumption, which indicates the consistency between the local optimal actions and the optimal global joint action. VDN (Lowe et al., 2017) and QMIX (Rashid et al., 2018) introduce additivity and monotonicity to Q-functions. QTRAN (Son et al., 2019) transforms IGM into optimization constraints. QPLEX

(Wang et al., 2020a) uses duplex dueling network architecture to guarantee IGM assumption. Instead of focusing on value decomposition, multi-agent policy gradient algorithms provide a centralized value function to evaluate current joint policy and guide the update of each local utility network. Most policy-based MARL methods extend single-agent RL ideas, including MADDPG (Lowe et al., 2017), MAPPO (Yu et al., 2022). FOP (Zhang et al., 2021) algorithm factorizes optimal joint policy by maximum entropy and MACPF (Wang et al., 2023) mixes critic values of each agent.

**TD($\lambda$) in Reinforcement Learning:** Recent works on TD($\lambda$) in MARL have explored the application and enhancement of TD($\lambda$), addressing the challenges in centralized value functions and policy gradients. SMIX($\lambda$) (Yao et al., 2021) uses an off-policy training to achieve a stable centralized value function by avoiding the greedy assumption and connects the SMIX($\lambda$) to Q($\lambda$) (Peng & Williams, 1994). ETD($\lambda$) (Jiang et al., 2021) ensures the convergence in the linear case by appropriately weighting TD($\lambda$) updates. Wang et al. (2020c) explores off-policy multi-agent learning with decomposed policy gradients, incorporating TD($\lambda$) methods for estimating the decomposed critic. Li et al. introduces a $\lambda$ annealing mechanism and a $\lambda^*$ threshold according to the training episodes. Importance sampling is the simplest way to correct for the discrepancy between behavior policy and target policy (Precup, 2000; Geist et al., 2014), but suffers from large variance. Retrace(Munos et al., 2016) is an off-policy reinforcement learning algorithm that uses truncated importance sampling with eligibility traces to enable safe and efficient value function updates from off-policy data. $Q^*(\lambda)$(Harutyunyan et al., 2016) introduces an off-policy correction based on the Q-baseline which avoids the blow-up of the variance but does not guarantee convergence for arbitrary $\pi$ and $\mu$. Tree-backup algorithm (Precup, 2000) corrects the discrepancy by multiplying each term of the sum of the product of target policy probabilities, however, it is not efficient in the near on-policy case as it unnecessarily cuts the traces. Motivated by (Hu et al., 2021), our method introduces a calculation method based on the likelihood that the sampled transitions occur in the current policy to determine the $\lambda$ values during the training process instead of preset a hyper-parameter ahead of the training process.

## 3 BACKGROUND

**MARL modeling** A fully cooperative multi-agent task is described as a Dec-POMDP (Oliehoek et al., 2016) task which consists of a tuple $G = \langle S, A, P, r, Z, O, N, \gamma \rangle$ in which $s \in S$ is the global state of the environment and $N$ is the number of agents. At each time step, each agent $i \in N \equiv \{1, \ldots, n\}$ chooses an action $a_i \in A$ which forms the joint action $\mathbf{a} \in \mathbf{A} \equiv A^N$. The transition on the environment is according to the state transition function that $P(\cdot|s, \mathbf{a}) : S \times \mathbf{A} \times S \to [0, 1]$. The reward function, $r(s, \mathbf{a}) : S \times A \to \mathbb{R}$, is shared among all the agents, and $\gamma \in [0, 1)$ is the discount factor for future reward penalty. Partially observable scenarios are considered in this paper that each agent draws individual observations $z_i \in Z$ of the environment according to the observation functions $O(s, i) : S \times N \to Z$. Meanwhile, the action-observation history, $\tau_i \in H \equiv (Z \times A)^*$, is preserved for each agent and conditions the stochastic policy $\pi_i(a_i|\tau_i) : H \times A \to [0, 1]$. In the Centralized Training with Decentralized Execution (CTDE) settings, the state is provided during the centralized training phase and the agents can only acquire partial observations during the decentralized execution phase.

**MARL algorithms** Value-based MARL algorithm aims to find the optimal joint action-value function $Q^*(s, \mathbf{a}; \theta) = r(s, \mathbf{a}) + \gamma \mathbb{E}_{s'}[\max_{\mathbf{a}'} Q^*(s', \mathbf{a}'; \theta)]$ and parameters $\theta$ are learned by minimizing the expected TD error. VDN learns a joint action-value function $Q_{tot}(\tau, \mathbf{a})$ as the sum of individual value functions: $Q_{tot}^{\text{VDN}}(\tau, \mathbf{a}) = \sum_{i=1}^{n} Q_i(\tau_i, a_i)$. QMIX introduces a monotonic restriction $\forall i \in N, \frac{\partial Q_{tot}^{\text{QMIX}}(\tau, \mathbf{a})}{\partial Q_i(\tau_i, a_i)} > 0$ to the mixing network to meet the IGM assumption. In policy-based algorithms, agents use a policy $\pi_\theta(a_i|\tau_i)$ parameterized by $\theta$ to produce an action $a_i$ from the local observation and jointly optimize the discounted accumulated reward $J(\theta) = \mathbb{E}_{a^t, s^t}[\sum_t \gamma^t r(s^t, a^t)]$ where $a^t$ is the joint action at time step $t$. In the AC-based algorithm, MAPPO algorithm, the actor is updated by optimizing the target function $J_{\theta^k}(\theta) = \sum_{s^t, a^t} \min(\frac{\pi_\theta(a^t|s^t)}{\pi_{\theta^k}(a^t|s^t)} A_{\theta^k}(s^t, a^t), clip(\frac{\pi_\theta(a^t|s^t)}{\pi_{\theta^k}(a^t|s^t)}, 1 - \epsilon, 1 + \epsilon) A_{\theta^k}(s^t, a^t))$, where the $\epsilon$ is the clip parameter and $A_{\theta^k}(s^t, a^t)$ is the advantage function. The critic training is similar to value-based $Q$ learning by calculating TD-error and TD targets. During the TD training process, the target value is calculated by bootstrapping from the existing Q-function according to temporal difference methods or Monte-Carlo returns. Temporal-difference algorithms are based on the fact that the value function

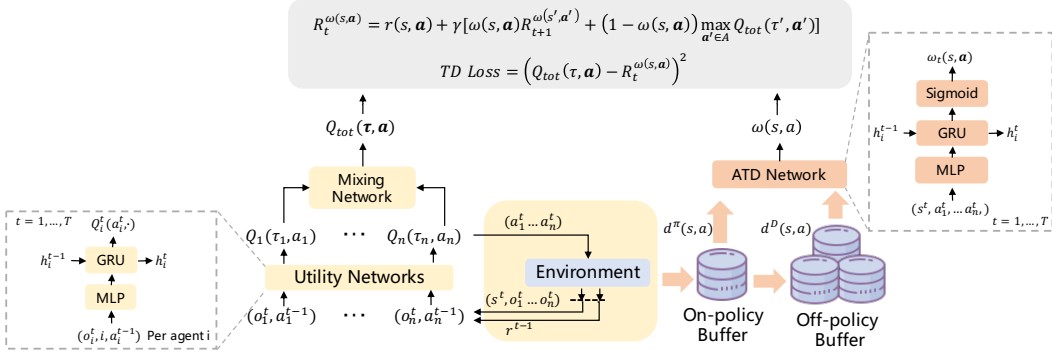

Figure 2: The utility networks and the mixing network are from the original MARL algorithms, QMIX in this paper. Interactive transitions are stored in two replay buffers. One of them is small (on-policy buffer) and the other one is large (off-policy buffer). Training data are sampled uniformly from these two buffers and used for calculating the likelihood-free density ratios. The density ratios are used as the $\lambda$ values and importance weights during the training process.

should satisfy Bellman equations for all $s$ and the target can be formulated using the regression target:

$$T_{TD(0)}(s_t, \mathbf{a_t}) := r(s_t, \mathbf{a_t}) + \gamma \hat{Q}(s_{t+1}, \mathbf{a_{t+1}}) \tag{1}$$

in which the $T$ is the target value, the $\hat{Q}(s_{t+1}, \mathbf{a_{t+1}})$ is the estimated Q value in $t+1$ time step and these algorithms are referred as TD(0). The Monte-Carlo approach is based on the intuition that the discounted sum of rewards realized by the policy from a state $s_t$ is an unbiased estimator of $Q(s_t, \mathbf{a_t})$. The target value is calculated by:

$$T_{MC}(s_t, \mathbf{a_t}) = \sum_{k=0}^{n_i-t-1} \gamma^t r(s_{t+k}, \mathbf{a_{t+k}}) \tag{2}$$

where the $n_i$ is the length of the trajectory $\tau_i$ and these algorithms are also referred as TD(1).

**TD($\lambda$)**    Between the two extremes of TD(0) and TD(1), TD($\lambda$) integrates the Temporal Difference method with Monte-Carlo methods by parameter $\lambda$:

$$T_t^\lambda = r(s_t, \mathbf{a_t}) + \gamma[\lambda T_{t+1}^\lambda + (1-\lambda) \max_{\mathbf{a'} \in A} Q(s_{t+1}, \mathbf{a'})]. \tag{3}$$

The TD($\lambda$) calculation is also related to the $\lambda$-return extension (Sutton, 1988) which considers the exponentially weighted sum of $n$-step returns to calculate $Q$ values. The general return-based operator $\mathcal{R}$ (Munos et al., 2016) is defined as:

$$\mathcal{R}Q(s, \mathbf{a}) := Q(s, \mathbf{a}) + \mathbb{E}_\mu[\sum_{t \geq 0} \gamma^t (\prod_{i=0}^{t} c_i)(r_t + \gamma \mathbb{E}_\pi Q(s_{t+1}, .) - Q(s_t, \mathbf{a_t}))] \tag{4}$$

for some non-negative coefficients $c_i$, traces, where $\mathbb{E}_\pi Q(s, .) := \sum_{\mathbf{a}} \pi(\mathbf{a}|s)Q(s, \mathbf{a})$ and $c_0 = 1$.

According to the Bellman Equation, for a policy $\pi$, the Bellman operator $\mathcal{T}^\pi$ is defined as $\mathcal{T}^\pi Q := r + \gamma P^\pi Q$, and the Bellman optimality operator is $\mathcal{T}Q := r + \gamma \max_\pi P^\pi Q$, where the $P^\pi$ operator is defined as $P^\pi Q(s, \mathbf{a}) := \sum_{s' \in S} \sum_{\mathbf{a'} \in A} P(s'|s, \mathbf{a}) \pi(\mathbf{a'}|s') Q(s', \mathbf{a'})$. In the policy evaluation setting, a fixed policy $\pi$ is given whose value $Q^\pi$ we wish to estimate from sample trajectories drawn from the behavior policy $\mu$. In the rollout process, the policy depends on the sequences of Q-functions, such as $\epsilon$-greedy policies, and seeks to approximate $Q^*$. Based on the notations above, the calculation of the off-policyness measurement is shown in the Method and the convergency analysis of $\mathcal{R}$ operator is shown in the Appendix A.

## 4 METHOD

In this section, we introduce the overall architecture of our framework and the training details of the likelihood-free density ratio network. Our framework generates two replay buffers of different sizes,

a large buffer for off-policy trajectories and a small buffer for on-policy trajectories. The $\lambda$-predictor network is trained adversarially by the transitions sampled from the two buffers. The network output is then scaled and used as the $\lambda$ value to calculate the target Q values. Additionally, we provide the implications of the adaptive $\lambda$ method on the convergence guarantees of MARL algorithms.

### 4.1 $\lambda$ VALUE ASSIGNMENT

In value-based MARL algorithms with mixing networks, the training process involves the policy improvement process and the credit assignment process among the agents. At the early training process after parameter weights initialization of the networks, the main target of the training process is to make $Q_{tot}$ values closing in on the cumulative returns. After the stability of the $Q_{tot}$ values, the training process continues to focus on the credit assignment tasks to allocate more accurate Q values.

To show the importance of different $\lambda$ to the training process as well as the credit assignment process, we conduct an experiment on the Spread task with two agents from the petting-zoo (Terry et al., 2021) environment. The rewards are given according to the minus distance between the agents and targets. We show the difference between the real cumulative return and the predicted $Q_{tot}$ from the mixing network of the initial state and report the training curves of the $Q$ value of each agent. Due to the fast convergence speed in the easy scenario, we only set the maximum running time step as 300k.

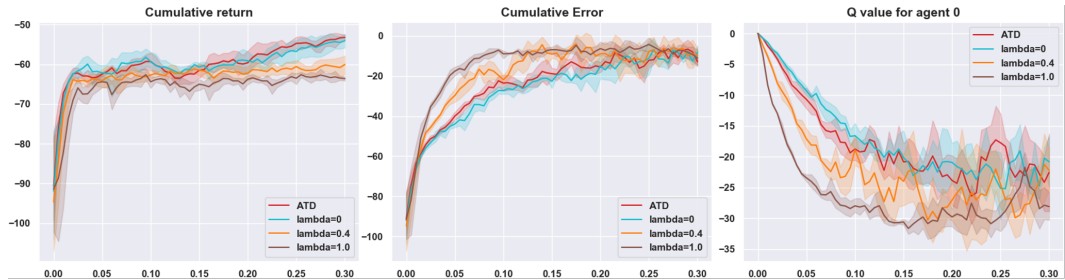

Figure 3: The difference between predicted $Q_{tot}$ values and the real returns and the two $Q$ values of each agent. The x-axis represents the time steps (1e6) being evaluated and the y-axis is the mean of the winning rate among 5 seeds with 32 evaluation processes.

According to the three graphs in Figure 3, the cumulative return indicates the final performance of different lambda values, and the cumulative error graph shows how accurately the networks can predict compared to the real return. In the early training steps, the predicted values are initial values and are slowly close to the real state-action values. The other two graphs show the initial $Q$ values of each agent and the amplitude of the changes shows the credit assignment process.

According to the cumulative error graph, a large $\lambda$ value makes the predicted $Q_{tot}$ quickly converge to the expected real returns. The target value from the TD-error calculation is from the real Monte-Carlo return but may suffer from the historical suboptimal trajectories. In contrast, the target value calculated with small $\lambda$ values is more in bootstrapping the previous predicted $Q_{tot}$ value from the network, which results in the slower convergence to the expected real return. Similarly, according to the graphs showing the $Q$ values of each agent, larger $\lambda$ values make the $Q_{tot}$ stable much faster and begin to concentrate on the credit assignment process because the values begin to vary at early time steps. In contrast, the values begin to change largely at late time steps when the $\lambda$ value is small because the $Q_{tot}$ stables lately.

Therefore, the $\lambda$ values influence the stability speeds of the mixing network and promote the credit assignment process in advance. Meanwhile, the $\lambda$ value should also trade-off between the training precision it takes with the training speed. In the next part, we will show empirically the effectiveness of our method compared with different preset $\lambda$ values.

### 4.2 ARCHITECTURE

The importance sampling between the behavior policy and the target policy according to the stability condition of the off-policy TD learning is currently the best method to adjust the data distribution (Sutton et al., 2016; Jiang et al., 2021). Meanwhile, given a fixed target policy $\pi$ and behavior policy $\mu$,

and a set of non-negative coefficients $c_t = \omega(\mathbf{a}_t, \tau_t)$ under the assumption that $0 \leq c_s \leq \frac{\pi(\mathbf{a}|s)}{\mu(\mathbf{a}|s)} \leq 1$, the use of importance sampling ,the operator $\mathcal{R}$ is $\gamma$-contraction (Munos et al., 2016). Moreover, the coefficients of Equation 4 are state-action specific, so that the coefficients can be represented by a parametric network conditioned on state-action pairs. Detail descriptions are in Appendix A.

The training process of value-based MARL algorithms is the $Q$ value Temporal Difference (TD) updating of each agent's utility network. In QMIX and the algorithms derived from QMIX, TD updates are applied to the mixed $Q_{tot}$ value. The utility network is composed of multi-layer perceptron (MLP) layers and Gate Recurrent Unit (GRU) cells in which $h_i^t$ is the historical hidden state. Similar to the QMIX algorithm, the utility network at time step $t$ of agent $i$ takes the observation $o_i^t$ and its chosen action $a_i^t$ as an input and outputs the $Q_i(\tau_i, a_i)$ of each agent according to the encoded history state $\tau_i$. Then, these $Q$ values are used for subsequent mixing mechanisms, QMIX and QPLEX for example, and trained by TD learning.

As shown in Figure 2, the interaction trajectories with the environments are stored in the original replay buffer (off-policy buffer) and a small on-policy buffer. The on-policy buffer is refreshed faster than the off-policy buffer, thus the transitions are more of an on-policy property. The new interactive trajectories are inserted into the fast buffer and the slow buffer. We then sample trajectories from both the replay buffers and train a network that takes state-action $(s, \mathbf{a})$ as input to calculate the on-policy density ratio. The ATD network is also composed of a multi-layer perceptron (MLP) layer and Gate Recurrent Unit (GRU) cells. The results are then activated by a sigmoid layer to be scaled in the range between $0$ and $1$. During the centralized training process, the global states are available so the recurrent layer can be masked because of the Markov property. As for the algorithms without centralized learning, the recurrent layer encodes the history observations and represents the latent state distribution. The ATD network takes the observation and the action of each agent as the input and provides the $\lambda$ values for each agent. Finally, the $\lambda$ values are used for calculating the eligibility trace and participate in the TD error calculation and the ATD network is updated with the same frequency as the target networks.

### 4.3 Measurement of on-policy transitions

We quantify the off-policy degree ($1-$ on-policy) of a transition based on its age, aligning with the rationale that transitions generated by older policies, denoting a higher off-policy nature, are not accurate via MC simulation and should be assigned gradually decreasing TD($\lambda$) values. The $\lambda$ value depends on the likelihood that an off-policy transition is generated by the current policy. We define $d$ as the distribution that the replay buffer $D$ is sampled from and is supported on the entire state-action space, $d^\pi$ as the stationary distribution of state-action pairs under the current policy, and $L_Q(\theta, d^\pi) = \|Q_{tot}(s, \mathbf{a}|\theta) - \mathcal{R}Q_{tot}(s, \mathbf{a}|\theta)\|_{d^\pi}^2$ as the loss function of the mixing network, in which the adaptive $\lambda$ value is calculated by $\omega(s, \mathbf{a})$.

In practice, obtaining an accurate estimation of $d^\pi$ requires on-policy samples from $d^\pi$ and interactions with the environment. Moreover, when incorporating off-policy transitions from the replay buffer, calculating the on-policy degree $\omega(s, \mathbf{a}) := d^\pi(s, \mathbf{a})/d^D(s, \mathbf{a})$ becomes challenging due to the replay buffer $D$ constituting a mixture of trajectories derived from policies at different time steps. In this paper, we adopt a variational representation of $f$-divergences between a set of older trajectories and a set of more recently generated trajectories to estimate the density ratios.

**Theorem 4.1** *(Nguyen et al., 2010) Assume that $f$ has first order derivatives $f'$ at $[0, +\infty)$. $\forall P, Q \in \mathcal{P}(\mathcal{X})$ such that $P \ll Q$ and $\omega : \mathcal{X} \to \mathbb{R}^+$,*

$$D_f(P\|Q) \geq \mathbb{E}_P[f'(\omega(x))] - \mathbb{E}_Q[f^*(f'(\omega(x)))] \tag{5}$$

*where $f^*$ denotes the convex conjugate and the equality is achieved when $\omega = \frac{dP}{dQ}$.*

According to Theorem 4.1, the density ratio $\omega(s, a) := d^\pi(s, a)/d^D(s, a)$ can be estimated by the samples from two sets of trajectories. One of the two sets of trajectories $d^D$ can be sampled from the regular large replay buffer (off-policy buffer $D_{off}$) from original value-based MARL algorithms and the other one $d^\pi$ from the small replay buffer (on-policy buffer $D_{on}$) which only contains recent trajectories. After each rollout process, the new trajectory is updated to $D_{on}$.

## 4.4 TRAINING PIPELINE

Based on the two samples from the off-policy replay buffer and the on-policy replay buffer, the $\omega(s, \mathbf{a})$ can be estimated by a network $\omega_\phi(s, \mathbf{a})$ parametrized by $\phi$. To estimate the density ratio $\omega(s, a) = \frac{dP}{dQ}(s, a)$, the problem can be framed as an optimization task of maximizing the lower bound based on the inequality. Maximizing the right-hand side of the inequality:

$$\max_{\omega_\phi}(\mathbb{E}_P[f'(\omega_\phi(s, \mathbf{a}))] - \mathbb{E}_Q[f^* f'(\omega_\phi(s, \mathbf{a}))]) \tag{6}$$

is equivalent to minimizing the negative lower bound of:

$$L_\omega(\phi) := \mathbb{E}_{D_{on}}[f^*(f'(\omega_\phi(s, \mathbf{a})))] - \mathbb{E}_{D_{off}}[f'(\omega_\phi(s, \mathbf{a}))] \tag{7}$$

From Theorem 4.1, the estimate of the density ratio can be recovered from the $\omega_\phi$ by minimizing the $L_\omega(\phi)$. Additionally, the output of the network $\omega_\phi$ is scaled to the range of $(0, 1)$ by the sigmoid activation function. The $f$ function chosen in this paper is symmetric divergence related to Jensen-Shannon (JS) divergence, which results in a binary cross-entropy loss:

$$L_\omega(\phi) := BCE_{s, \mathbf{a} \sim D_{off}}(\omega_\phi(s, \mathbf{a}), 0) + BCE_{s, \mathbf{a} \sim D_{on}}(\omega_\phi(s, \mathbf{a}), 1) \tag{8}$$

The detailed mathematical derivation is shown in Appendix A.4

Therefore, the final objective for TD learning over Q is then:

$$L_Q(\theta; d^\pi) \approx L_Q(\theta; D_{off}, \omega_\phi) = \mathbb{E}_{(s, \mathbf{a}) \sim D_{off}}[(Q_{tot}(s, \mathbf{a}|\theta) - R_t^{\omega(s, \mathbf{a})})^2] \tag{9}$$

$$R_t^{\omega(s, \mathbf{a})} = r(s, \mathbf{a}) + \gamma[\omega(s, \mathbf{a}) R_{t+1}^{\omega(s', \mathbf{a}')} + (1 - \omega(s, \mathbf{a})) \max_{\mathbf{a}' \in A} Q_{tot}(s', \mathbf{a}'|\theta^-)] \tag{10}$$

where the $\theta^-$ is the mixing network parameter which is maintained and updated frequently. The $R_t^{\omega(s, a)}$ in this formula is the target value at time step $t$ calculated by TD($\lambda$) in which the $\lambda$ value is calculated by $\omega$ network and conditioned on the state-action $(s, \mathbf{a})$ pairs.

In basic value-based MARL algorithms, the parameter of the utility network or the mixing network $\theta$ is updated frequently and the lag parameter $\theta^-$ is employed during the calculation of the target mixed value. Consequently, for a given state-action pair, the target value remains unchanged between update intervals, providing a stable supervised signal to $Q_{tot}(s, \mathbf{a}|\theta)$. Therefore, we cache the $\lambda$ value for each sampled transition based on $\omega(s, \mathbf{a})$ for subsequent use, and refresh these cached values to 0 after the update of $\theta^-$ to ensure the stability.

For MAPPO, the two buffers are also maintained. The first is the standard on-policy rollout buffer, whose capacity corresponds to the number of rollout threads defined by MAPPO, which serves as the original trajectory buffer. To incorporate additional historical experience, a second off-policy buffer with a size 50x larger than the on-policy buffer is introduced. In the standard MAPPO framework, the actor is optimized using GAE computed from the on-policy data, while the critic is trained by minimizing the mean-squared error between its value estimates and the cumulated (MC) return. In our modification, we replace the MC return with the ATD-based return, which is computed using samples drawn from both the on-policy and off-policy buffers. This changes the critic learning procedure accordingly while leaving the actor update unchanged, aside from the addition of the off-policy buffer used for ATD estimation.

## 5 EXPERIMENT

We evaluate the performance of our method via the fully cooperative StarCraftII micro-management challenges by the mean winning rate in each scenario and the average scoring results in Google Football Research. In this section, we mainly show the effectiveness of our ATD($\lambda$) method by comparing the dynamically assigned $\lambda$ values to those commonly used preset TD($\lambda$) values. We also show the performance enhancement by presenting 6 out of 23 scenarios with 2 levels of difficulty from SMAC in this paper and 6 multi-player academy tasks from Gfootball in Appendix C. Additionally, ablation studies are also conducted to show the adaptability of our approach to other algorithms and the influence of fast buffer size.

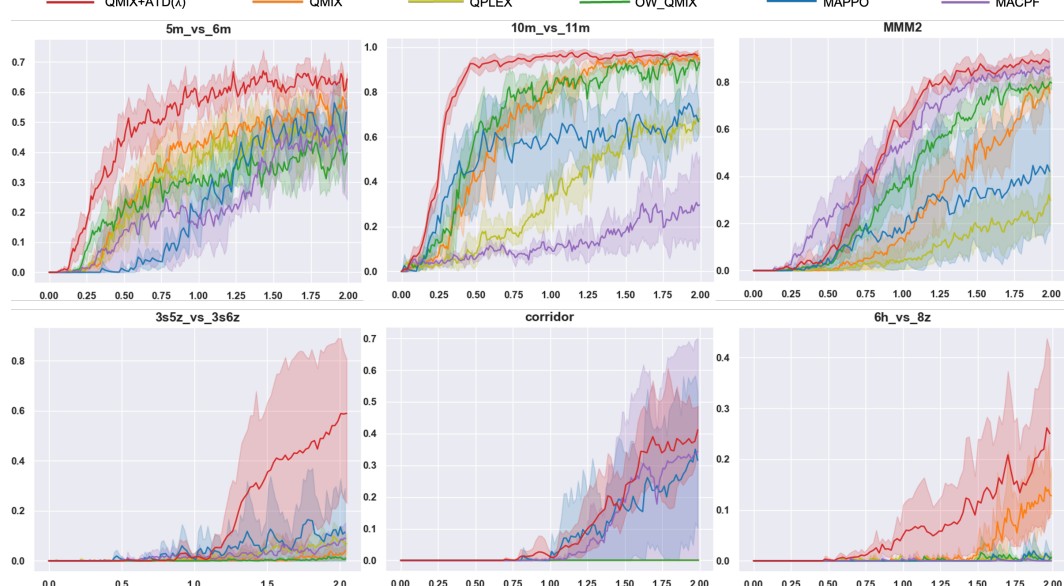

Figure 4: The winning rate curves evaluated on the 6 SMAC tasks with two major difficulties among our ATD+QMIX and other baseline algorithms.

## 5.1 EXPERIMENT SETTINGS

**SMAC** We verify our proposed adaptive $\lambda$ methods on 6 subtasks of two difficulties, **a)** hard tasks including 5m_vs_6m, 10m_vs_11m, and **b)** super-hard scenarios 3s5z_vs_3s6z, corridor, MMM2, and 6h_vs_8z. The difficulty is set as 7 by default. The winning rates of battles are calculated by the mean of 32 evaluation processes. We repeat the experiment 10 times with different seeds and smoothed by 0.6 for better visualization within 2M time steps. The shading area is the variance of the 10 different seeds and represents the stability of the generated policies. In each scenario from each experiment, the x-axis represents the time steps (e6) being evaluated and the y-axis is the mean of the winning rate among 5 seeds of 32 evaluation rollout rounds.

**Baseline** We adapt our method to QMIX and MAPPO and compare our methods to the value-based W-QMIX and QPLEX, popular policy-based algorithm MAPPO, and currently the latest AC-based algorithm MACPF with their officially-provided default parameter settings. The QMIX, QPLEX, and W-QMIX in this paper are from the pymarl codebase (Rashid et al., 2020). The MACPF is from the codebase (Zhang et al., 2021; Wang et al., 2023) and the MAPPO is provided by Yu et al. (2022).

## 5.2 EXPERIMENT RESULTS

In this section, we show the testing curves of our proposed adaptive $\lambda$ method on the QMIX algorithm with TD($\lambda$) across six benchmarks within the SMAC framework which encompass two hard tasks and four super-hard tasks. The average test winning rate, computed across 32 seeds 10 times for each of the 6 scenarios, is depicted in Figure 5 to provide a comprehensive overview of the algorithms' overall performance. According to the officially provided codebases of other baseline algorithms, different suggested $\lambda$ values are pre-defined in the config files. For instance, in the QMIX and QPLEX implementation in pymarl2 (Hu et al., 2021), the $\lambda$ value is set as 0.4. In the WQMIX algorithm, the value is set as 0.6 and 0.8 in the MACPF config. Therefore, we compare our adaptive TD($\lambda$) value with the popular commonly used values, including TD(0) for fully TD update, TD(1) for fully Monte-Carlo methods, direct importance sampling calculation and Retrace calculation.

According to Figure 5, the ATD($\lambda$) method outperforms other preset $\lambda$ values, direct importance sampling, and the Retrace algorithm in the three hard tasks and two super-hard tasks, and competes favorably with importance sampling in the corridor scenario. In other easy tasks, almost all the $\lambda$ value settings can achieve similar convergence performance. In the convergency analysis section, the traces meet the requirement $0 \leq c_s \leq \frac{\pi(\mathbf{a}|s)}{\mu(\mathbf{a}|s)} \leq 1$ so that the upper bound of $\lambda$ value conditioned on state-action pair is the importance sampling result. As a consequence, the $\lambda = 0$ is the most

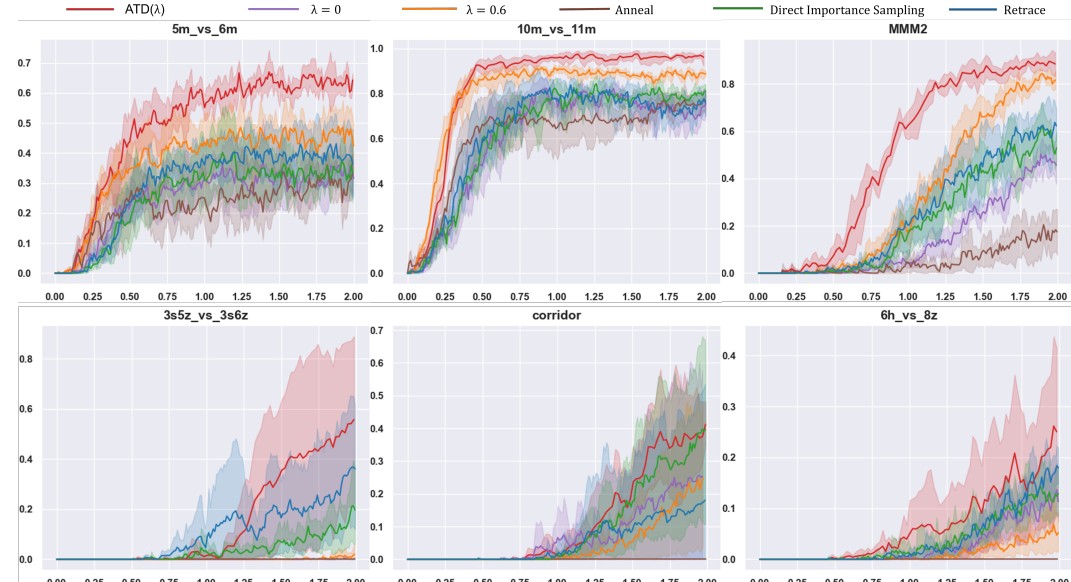

Figure 5: The winning rate curves evaluated on the 6 SMAC tasks with two major difficulties. The baseline algorithms are the QMIX with different commonly-used $\lambda$ values and three methods for adaptive $\lambda$ calculation methods.

conservative setting which guarantees the convergence if sufficient training time steps are given. Similarly, small $\lambda$ values have larger probabilities to stay in the range between $0 \leq c_s \leq \frac{\pi(\mathbf{a}|s)}{\mu(\mathbf{a}|s)}$, which provides acceptable convergence speed without small risks. In contrast, large pre-defined $\lambda$ values have probabilities of overflowing from the range, which results in large-biased target values and unstable performances in the provided graphs.

The paper (Hu et al., 2021) suggests a large $\lambda$ value to solve the super-hard tasks, which is caused by the parallel rollout runner essence. From the suggested hyper-parameter settings in the config files from the codebases, the $\lambda$ values are small in the episode rollout runner and large $\lambda$ values are recommended in the parallel runner. When utilizing the parallel runner during the rollout process, a number of new trajectories are sampled and inserted into the replay buffer. Thus, the replay buffer with limited size should contain the trajectories with less diversity and be more on-policy compared with that using episode runner. In such a way, the $\frac{\pi(\mathbf{a}|s)}{\mu(\mathbf{a}|s)}$ values are closer to 1 and result in large $\lambda$ values, which is consistent with the intuition provided by Hu et al. (2021).

## 5.3 DISCUSSION

In this section, we mainly show the performance enhancement and the compatibility that our method can be adapted to other value-based methods. We also show the influence of on/off-policy buffer size and the $\lambda$ value cache mechanism.

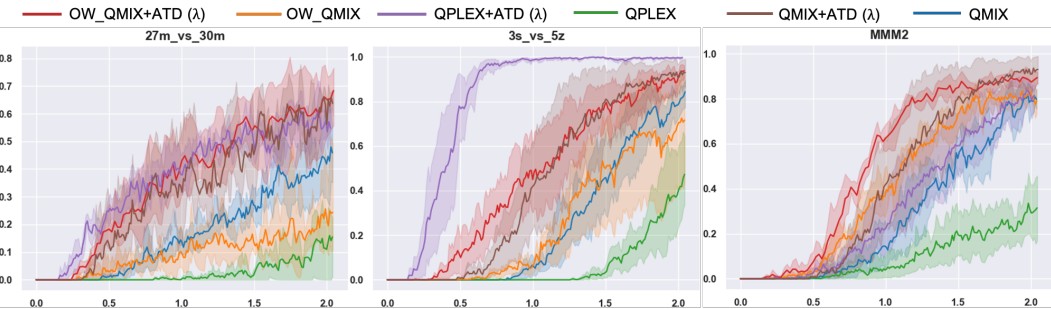

Figure 6: The winning rate curves evaluated on 27m_vs_30m, 3s_vs_5z, and MMM2 scenarios.

**Performance Enhancement**   We commence the evaluation of our proposed adaptive $\lambda$ method on the QMIX algorithm with TD($\lambda$) across six benchmarks within the SMAC framework which encompass two hard tasks and four super-hard tasks. The average test winning rate, computed across 10 seeds for each of the 6 scenarios, is depicted in Figure 4 to provide a comprehensive overview of the algorithms' overall performance. In hard tasks such as 5m_vs_6m and 10m_vs_11m, our proposed method competes favorably with or outperforms other baseline algorithms. In the 3s5z_vs_3s6z, 6h_vs_8z, MMM2, and the corridor task, where not all baseline algorithms exhibit winning rates, our method achieves commendable results. Notably, for the 3s5z_vs_3s6z task, we fine-tune the parameter of the mixing network size in QMIX and apply both the original setting and the adjusted setting to other baselines. The graph reflects the superior performance of the two settings. Other hyper-parameters are detailed in Appendix E.

**Compatibility**   We implement two replay buffers of different sizes and calculate the $\lambda$ value according to the state-action density ratios, so our ATD($\lambda$) module, based on TD updates, can be regarded as a plugin that can be adapted to other value-based MARL methods with minor changes. To test the compatibility of our work, we apply our method on QPLEX, and OW_QMIX algorithms in 27m_vs_30m, 3s_vs_5z, and MMM2 scenarios corresponding-ingly.

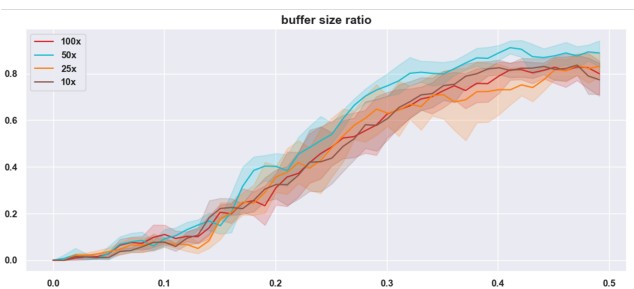

Figure 7: The winning rate curves evaluated on 10m_vs_11m with different ratios between the on/off-policy replay buffer size.

According to Figure 6, in the three scenarios, all of the algorithms with our proposed ATD($\lambda$) method outperform the algorithms with static $\lambda$ values. By adding a large replay buffer and updating the target critic value, we also apply the idea of our approach to the MAPPO algorithm to assist the critic network training. Figure 9 in the Appendix C shows that our method can also improve the policy-based MARL algorithms with critic networks.

**On/Off-policy Buffer Size**   According to the default settings of code-base pymarl, we set the size of the off-policy replay buffer as 5000. To test the influence of the on-policy replay buffer size, we choose four ratios, 10x, 25x, 50x, and 100x, compared with the off-policy replay buffer on the 10m_vs_11m scenario. As for the MAPPO algorithm, the on-policy buffer size is the parallel rollout size and the off-policy buffer size is calculated by multiplying the ratios.

As shown in Figure 7, the 50x buffer size distinctly achieves the highest performance and faster convergence speed. Empirically with the policy improvement progress, a large on-policy buffer may be mixed into some off-policy data because old trajectories are refreshed slowly. In contrast, a small on-policy buffer may not be able to contain enough on-policy data due to the variance initial state. Thus, this paper chose a proper ratio of 50x and used it as the default setting among the experiments.

## 6   CONCLUSION AND FUTURE WORK

In this work, we consider the challenge of choosing the hyperparameter of the $\lambda$ value, which should be properly selected before training. We propose our ATD method that consists of two replay buffers and an extra network to calculate the $f$-divergence as the density ratio. Most MARL code bases have either provided the off-policy buffer (value-based methods) or on-policy buffer (policy-based methods), so our ATD can be easily implemented. The SMAC and Gfootball experimental results indicate that our method can be adapted to both value-based and AC-based methods. However, our method may not be suitable to the environment with too much randomness, the transitions in the replay buffer are non-repeating, such that the density ratio, $\omega$, are always quickly decayed to $0$. More discussions about SMACv2 are in Appendix D. In the future, we might be concerned about how to solve the problem taken by large randomness such as the SMACv2 environment.

## 7 CHECKLIST

### 7.1 DECLARATION OF LLM USAGE

During the paper writing, LLMs are used solely for polishing the writing, such as correcting spelling and grammar errors, and for no further purpose.

### 7.2 ETHICS

There are no ethical concerns currently because the codebase, the environment, and the data are open-sourced and are cited in the paper.

### 7.3 REPRODUCIBILITY

The testbed is publicly accessible from GitHub, and StarCraft II is provided by Storm platform. The codes are also provided in the supplementary materials.

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

# A  ANALYSIS OF ATD($\lambda$)

In this section, we first introduce the necessity of importance sampling between the behavior policy and the target policy according to the stability condition of off-policy TD Learning (Sutton et al., 2016; Jiang et al., 2021). Then, we show that given a fixed target policy $\pi$ and behavior policy $\mu$, and a set of non-negative coefficients $c_t = \omega(\mathbf{a}_t, \tau_t)$ under the assumption that $0 \leq c_s \leq \frac{\pi(\mathbf{a}|s)}{\mu(\mathbf{a}|s)} \leq 1$, the use of importance sampling ,the operator $\mathcal{R}$ is $\gamma$-contraction (Munos et al., 2016). Finally, we mention that the coefficients from Equation 4 are state-action specific so that the coefficients can be represented by a parametric network conditioned on state-action pairs.

## A.1  NECESSITY OF IMPORTANCE SAMPLING

Sutton et al. (2016) introduced the stability condition based on the simplest function approximation case, that of linear TD(0) and constant discount-rate $\gamma \in [0, 1)$. Given a transition $(s_t, a_t, r_t, s_{t+1})$, the conventional linear TD(0) is the update to the parameter vector $\theta$:

$$
\begin{aligned}
\theta_{t+1} &= \theta_t + \alpha \left(r_t + \gamma V_{\theta_t}(s_{t+1}) - V_{\theta_t}(s_t)\right) \frac{\partial V_{\theta_t}(s_t)}{\partial \theta_t} \\
&= \theta_t + \alpha \left(r_t + \gamma \theta_t^\top \psi(s_{t+1}) - \theta_t^\top \psi(s_t)\right) \psi(s_t) \\
&= \theta_t + \alpha \left(r_t \psi(s_t) - \psi(s_t)(\psi(s_t) - \gamma \psi(s_{t+1}))^\top \theta_t\right) \\
&= (I - \alpha A_t)\theta_t + \alpha b_t
\end{aligned}
\tag{11}
$$

where $\alpha > 0$ is a step-size parameter and $\psi(s) \in \mathbb{R}^n$ is the feature vector corresponding to state $s$. The matrix $A_t$ multiplies the parameter $\theta_t$ and is thereby critical to the stability of the iteration. Meanwhile, Sutton et al. (2016) established the stability by proving that matrix $A$ is positive definite and the:

$$
\begin{aligned}
A = \lim_{t \to \infty} \mathbb{E}[A_t] &= \lim_{t \to \infty} \mathbb{E}_\pi[\psi(s_t)(\psi(s_t) - \gamma \psi(s_{t+1}))^\top] \\
&= \sum_s d_\pi(s)\psi(s) \left(\psi(s) - \gamma \sum_{s'} [P_\pi]_{ss'}\psi(s')\right)^\top \\
&= \Psi^\top D_\pi (I - \gamma P_\pi)\Psi
\end{aligned}
\tag{12}
$$

where the $\Psi$ is the matrix with the $\psi(s)$ as its rows, the $D_\pi$ is the diagonal matrix with $d_\pi$ on its diagonal, and the $P_\pi$ denotes transition probabilities matrix $[P_\pi]_{ij} = \sum_a \pi(a|i)p(j|i,a)$. The on-policy learning process is stable because both the data distribution and the transition probability are based on the same policy $\pi$. However, the $A$ matrix for off-policy learning is:

$$
A = \Psi^\top D_\mu (I - \gamma P_\pi)\Psi
\tag{13}
$$

where $D_\mu$ is the diagonal matrix with the stationary distribution $d_\mu$ on its diagonal. The distribution and the transition probabilities do not match, $P_\pi^\top d_\mu \neq d_\mu$, and the positive definite cannot be guaranteed. Therefore, the importance sampling method which connects the behavior policy $\mu$ and target policy $\pi$ distributions is necessary to maintain the stability of off-policy learning methods.

## A.2  CONVERGENCY ANALYSIS

Munos et al. (2016) provided the convergency analysis based on the importance sampling. Given a fixed target policy $\pi$ and behavior policy $\mu$, and a set of non-negative coefficients $c_t = \omega(\mathbf{a}_t, \tau_t)$ under the assumption that $0 \leq c_s \leq \frac{\pi(\mathbf{a}|s)}{\mu(\mathbf{a}|s)} \leq 1$, the operator $\mathcal{R}$ is $\gamma$-contraction. The Equation 4 can

be rewritten as:

$$\mathcal{R}Q(s,\mathbf{a}) := Q(s,\mathbf{a}) + \mathbb{E}_\mu[\sum_{t\geq0}\gamma^t(\prod_{i=0}^{t}c_i)(r_t + \gamma\mathbb{E}_\pi Q(s_{t+1},.) - Q(s_t,\mathbf{a}_t))]$$

$$= \sum_{t=0}\gamma^{t=0}[(\prod_{i=0}^{t=0}c_s)Q(s_{t=0},a_{t=0})]+$$

$$\mathbb{E}_\mu[\sum_{t\geq0}\gamma^t(\prod_{i=0}^{t}c_i)(r_t + \gamma\mathbb{E}_\pi Q(s_{t+1},.))] - \mathbb{E}_\mu[\sum_{t\geq0}\gamma^t(\prod_{i=0}^{t}c_i)Q(s_t,\mathbf{a}_t)] \quad (14)$$

$$= \sum_{t\geq0}\gamma^t\mathbb{E}_\mu[(\prod_{i=0}^{t}c_i)(r_t + \gamma\mathbb{E}_\pi Q(s_{t+1},.))] - \sum_{t\geq0}\gamma^{t+1}\mathbb{E}_\mu[(\prod_{i=0}^{t}c_i)c_{t+1}Q(s_{t+1},\mathbf{a_{t+1}})]$$

$$= \sum_{t\geq0}\gamma^t\mathbb{E}_\mu[(\prod_{i=0}^{t}c_i)(r_t + \gamma[\mathbb{E}_\pi Q(s_{t+1},.) - c_{t+1}Q(s_{t+1},\mathbf{a_{t+1}}))]].$$

According to the Bellman equation, $Q^\pi$ is the fixed point of $\mathcal{T}^\pi$. The $Q^\pi$ is also a fixed point of the operator $\mathcal{R}$ because $\mathbb{E}_{s_{t+1}\sim P(.|s_t,\mathbf{a}_t)}[r_t + \gamma\mathbb{E}_\pi Q^\pi(s_{t+1},.) - Q^\pi(s_t,\mathbf{a}_t)] = \mathcal{T}^\pi Q^\pi - Q^\pi(s_t,\mathbf{a}_t) = 0$. Therefore, defining $\Delta Q := Q - Q^\pi$, the the difference between $\mathcal{R}Q$ and its fixed point $Q^\pi$ is:

$$\mathcal{R}Q(s,\mathbf{a}) - Q^\pi(s,\mathbf{a}) = \mathbb{E}_\mu[\sum_{t\geq1}\gamma^t(\prod_{i=0}^{t-1}c_i)([\mathbb{E}_\pi[Q - Q^\pi(s_t,.)] - c_t(Q-Q^\pi)(s_t,\mathbf{a}_t)])]$$

$$= \sum_{t\geq1}\gamma^t\mathbb{E}_{s_{1:t}\mathbf{a}_{1:t-1}}[(\prod_{i=0}^{t-1}c_i)\sum_b(\pi(\mathbf{b}|s_t) - \mu(\mathbf{b}|s_t)c_t(\mathbf{b},\tau_t))\Delta Q(s_t,\mathbf{b})]. \quad (15)$$

Since $0 \leq c_t \leq \frac{\pi(\mathbf{a}|s)}{\mu(\mathbf{a}|s)} \leq 1$ and $\pi(\mathbf{b}|s_t) - \mu(\mathbf{b}|s_t)c_t(\mathbf{b},\tau_t) \geq 0$, the $\mathcal{R}Q - Q^\pi$ is a linear combination of non-negative coefficients weights $\Delta Q(s_t,\mathbf{b})$, which is $\mathcal{R}Q(s,\mathbf{a}) - Q^\pi(s,\mathbf{a}) = \sum_{y,\mathbf{b}}\omega_{y,\mathbf{b}}\Delta Q(s_t,\mathbf{b})$, where

$$\omega_{y,\mathbf{b}} := \sum_{t\geq1}\gamma^t\mathbb{E}_{s_{1:t}\mathbf{a}_{1:t-1}}[(\prod_{i=0}^{t-1}c_i)(\pi(\mathbf{b}|s_t) - \mu(\mathbf{b}|s_t)c_t(\mathbf{b},\tau_t))\mathbb{I}(s_t = y)].$$

The sum of these coefficients is:

$$\sum_{y,\mathbf{b}}\omega_{y,\mathbf{b}} = \sum_{t\geq1}\gamma^t\mathbb{E}_{s_{1:t}\mathbf{a}_{1:t-1}}[(\prod_{i=0}^{t-1}c_i)\sum_\mathbf{b}(\pi(\mathbf{b}|s_t) - \mu(\mathbf{b}|s_t)c_t(\mathbf{b},\tau_t))]$$

$$= \mathbb{E}_\mu[\sum_{t\geq1}\gamma^t(\prod_{i=0}^{t-1}c_i) - \sum_{t\geq1}\gamma^t(\prod_{i=0}^{t}c_i)] = \gamma C - (C-1) \quad (16)$$

where $C = \mathbb{E}_\mu[\sum_{t\geq0}\gamma^t(\prod_{i=0}^{t}c_i)] \geq 1$ ($\prod_{i=0}^{0}c_i = 1$) and $\sum_{y,\mathbf{b}}\omega_{y,\mathbf{b}} \leq \gamma$. Therefore, $\mathcal{R}$ is a $\gamma$-contraction mapping around $Q^\pi$.

**Coefficient Representation** It is worth mentioning that the coefficients $c$ depend on the history-action pairs in the above equations. Because of the Markov property, the history $\tau$ can be replaced by state $s$ during the centralized training process. As for the fully decentralized training process, where the encoding of history observations represents the latent state distribution (Venkatraman et al., 2017), the history-action pairs can also be used for training.

### A.3 PROOF OF THEOREM 4.1

**Theorem A.1** *Assume that $f$ has first-order derivatives $f'$ on $[0, +\infty)$. For all probability measures $P, Q \in \mathcal{P}(\mathcal{X})$ such that $P \ll Q$ and any non-negative function $\omega : \mathcal{X} \to \mathbb{R}^+$, the following inequality holds:*

$$D_f(P\|Q) \geq \mathbb{E}_P[f'(\omega(x))] - \mathbb{E}_Q[f^*(f'(\omega(x)))], \tag{17}$$

*where $f^*$ is the convex conjugate of $f$. Equality is achieved when $\omega = \frac{dP}{dQ}$.*

The proof relies on properties of the convex conjugate $f^*$ and the definition of $f$-divergence.

**Convex Conjugate Inequality:** By the definition of the convex conjugate $f^*$, for any $x \geq 0$ and $y \in \mathbb{R}$, we have:

$$f^*(y) \geq xy - f(x), \tag{18}$$

with equality if and only if $y = f'(x)$. Rearrange the formula and gives:

$$f(x) \geq xy - f^*(y). \tag{19}$$

**Apply to $\frac{dP}{dQ}$:** Let $x = \frac{dP}{dQ}(x)$ and $y = f'(\omega(x))$. Substituting into the inequality:

$$f\left(\frac{dP}{dQ}(x)\right) \geq \frac{dP}{dQ}(x) \cdot f'(\omega(x)) - f^*(f'(\omega(x))). \tag{20}$$

**Take Expectations with Respect to $Q$:** Integrate both sides with respect to $Q$:

$$\mathbb{E}_Q\left[f\left(\frac{dP}{dQ}\right)\right] \geq \mathbb{E}_Q\left[\frac{dP}{dQ} \cdot f'(\omega(x))\right] - \mathbb{E}_Q\left[f^*(f'(\omega(x)))\right]. \tag{21}$$

Then, The left-hand side is the $f$-divergence $D_f(P\|Q)$ and the first term on the right-hand side simplifies to $\mathbb{E}_P[f'(\omega(x))]$ because $\mathbb{E}_Q\left[\frac{dP}{dQ} \cdot g\right] = \mathbb{E}_P[g]$. Thus, we obtain:

$$D_f(P\|Q) \geq \mathbb{E}_P[f'(\omega(x))] - \mathbb{E}_Q[f^*(f'(\omega(x)))]. \tag{22}$$

**Equality Condition:** Equality holds in the conjugate inequality when $y = f'(x)$, meaning:

$$f'(\omega(x)) = f'\left(\frac{dP}{dQ}(x)\right). \tag{23}$$

If $f$ is strictly convex, $f'$ is injective, and thus:

$$\omega(x) = \frac{dP}{dQ}(x). \tag{24}$$

Therefore, the inequality becomes an equality if and only if $\omega = \frac{dP}{dQ}$.

### A.4 BCE LOSS FORMULATION

In this work, we use the Jensen-Shannon Divergence $f(x) = x \log x + (1 - x) \log(1 - x)$ as the $f$ function and the derivative of $f(x)$ is:

$$f'(x) = \frac{d}{dx}\left[x \log x + (1 - x) \log(1 - x)\right] = \log\left(\frac{x}{1 - x}\right). \tag{25}$$

Then, the convex conjugate is defined as $f^*(y) = \sup_x \left[xy - f(x)\right]$. To find the supremum, we set $\frac{d}{dx}\left[xy - f(x)\right] = y - f'(x) = 0 \implies y = f'(x) = \log\left(\frac{x}{1-x}\right)$. Based on that, solving for $x$ is:

$$x = \frac{e^y}{1 + e^y} = \sigma(y), \tag{26}$$

where $\sigma(y)$ is the sigmoid function.

Substituting back to the convex conjugate function:

$$f^*(y) = \sigma(y)y - f(\sigma(y))$$

$$= \sigma(y)\log\left(\frac{\sigma(y)}{1-\sigma(y)}\right) - [\sigma(y)\log\sigma(y) + (1-\sigma(y))\log(1-\sigma(y))] \qquad (27)$$

$$= -\log(1-\sigma(y)).$$

Since $\sigma(f'(x)) = x$, we have:

$$f^*(f'(x)) = -\log(1-x). \qquad (28)$$

Next, we substitute the results

$$f'(x) = \log\left(\frac{x}{1-x}\right),$$

$$f^*(f'(x)) = -\log(1-x),$$

to the original loss:

$$L_\omega(\phi) = \mathbb{E}_{D_{on}}\left[f^*(f'(x))\right] - \mathbb{E}_{D_{off}}\left[f'(x)\right].$$

Thus:

$$L_\omega(\phi) = \mathbb{E}_{D_{on}}\left[-\log(1-x)\right] - \mathbb{E}_{D_{off}}\left[\log\left(\frac{x}{1-x}\right)\right]$$

$$= \mathbb{E}_{D_{on}}\left[-\log(1-x)\right] + \mathbb{E}_{D_{off}}\left[-\log(x) + \log(1-x)\right]. \qquad (29)$$

Assuming balanced expectations that $\mathbb{E}_{D_{off}}[\log(1-x)]$ cancels with $\mathbb{E}_{D_{on}}[-\log(1-x)]$, this reduces to:

$$L_\omega(\phi) = \mathbb{E}_{D_{off}}\left[-\log(x)\right] + \mathbb{E}_{D_{on}}\left[-\log(1-x)\right]$$

$$= \mathbb{E}_{D_{off}}\left[-1*\log(x) - (1-1)\log(1-x)\right] + \mathbb{E}_{D_{on}}\left[0*\log(x) - (1-0)\log(1-x)\right]$$

$$= BCE_{D_{off}}(x,0) + BCE_{D_{on}}(x,1)$$

$$(30)$$

## B  COMMONLY-USED $\lambda$ VALUES

The content in the main paper demonstrates the comparison among our ATD+QMIX with other commonly-used $\lambda$ values. In this subsection, we show all the learning curves from TD learning to Monte-Carlo learning process in Figure 8. In the main paper, we select three learning curves of lambda settings from this figure and add two other baseline algorithms. As shown in this graph, our proposed ATD($\lambda$) method can achieve the highest performance.

## C  GOOGLE FOOTBALL RESEARCH RESULTS

**Experiment Setting**  We also verify our proposed adaptive $\lambda$ method on 6 academy scenarios of Google Football Research of which the reward settings are sparse. Since value-based MARL algorithms underperform on sparse reward settings, we mainly show the improvement of adaptive $\lambda$ on critic training of AC-based methods within 50M time steps. The winning rates are also calculated by the mean of 32 evaluation processes. We repeat the experiment 10 times with different seeds and smoothed by 0.6 and the shading area is the variance of the 10 different seeds and represents the stability of the generated policies.

**Discussion**  Figure 9 presents the results of our proposed adaptive $\lambda$ method applied to the critic training of MAPPO algorithm across six academy scenarios within the Google Football Research framework. In the scenarios 'pass_shoot_with_keeper' (abbreviated as psk) and '3_vs_1_with_keeper' (3_vs_1k), our method competes effectively with the MAPPO algorithm. Impressively, in the remaining four scenarios, our method significantly outperforms other baseline algorithms. It is noteworthy that Figure 9 also highlights a trend where the majority of value-based algorithms and an

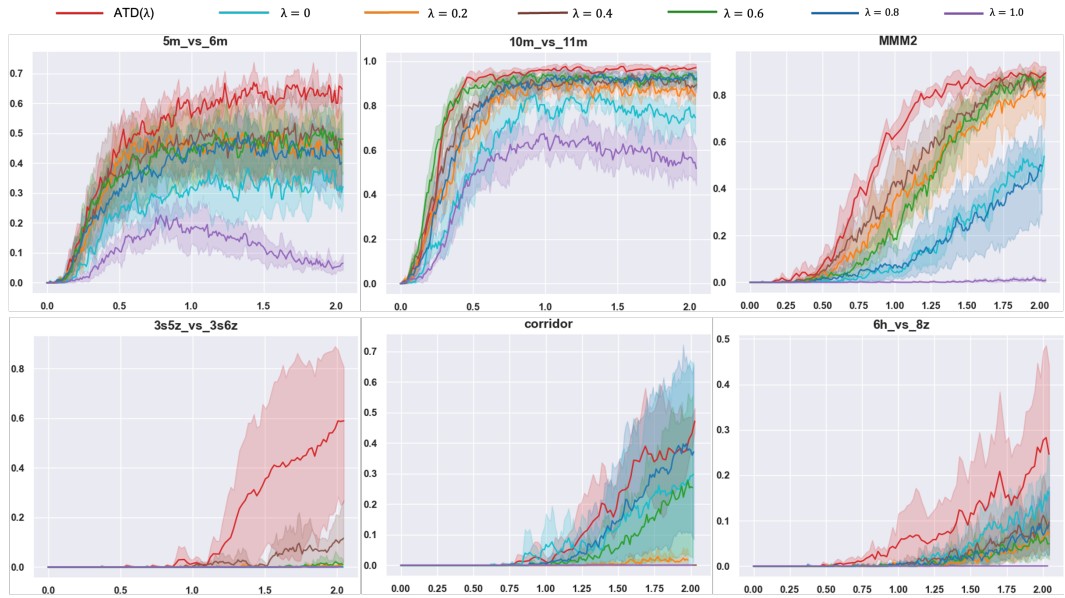

Figure 8: The winning rate curves evaluated on the 6 SMAC tasks with two major difficulties. The baseline algorithms are the QMIX with different commonly-used $\lambda$ values

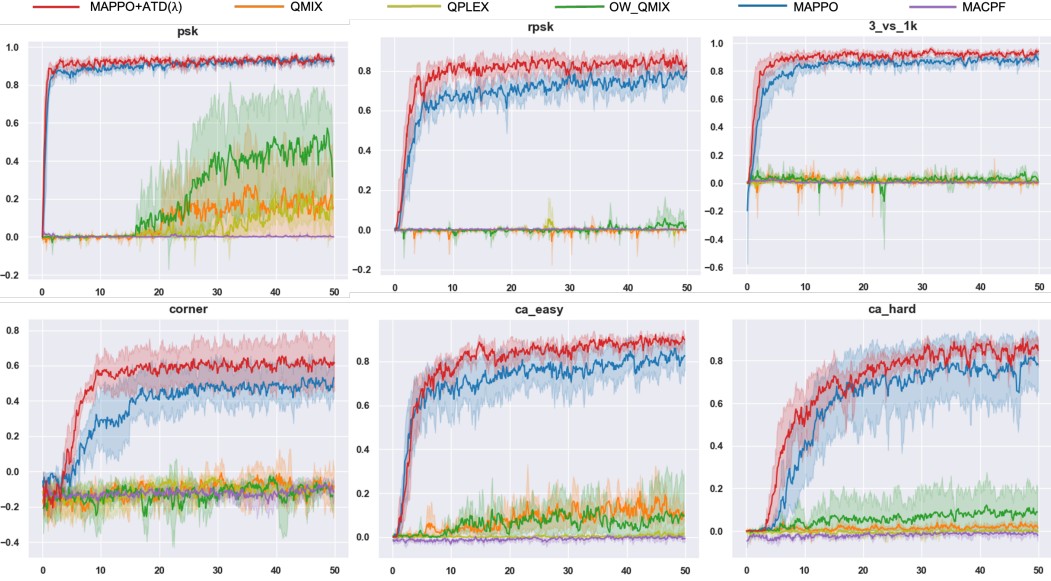

Figure 9: The winning rate curves evaluated on the 6 academies Gfootball tasks. The x-axis represents the time steps (1e6) being evaluated and the y-axis is the mean of the scores among 5 seeds.

Table 1: Performance in Gfootball tasks within 50M time steps

| Task | QMIX | QPLEX | OW-QMIX | MACPF | MAPPO | ATD-MAPPO |
|---|---|---|---|---|---|---|
| rpsk | 0.0008 | 0.0002 | 0.0234 | 0.0010 | 0.7846 | **0.8482** |
| psk | 0.1641 | 0.1707 | 0.4457 | 0.0001 | **0.9471** | 0.9312 |
| 3v1_k | 0.0025 | 0.0205 | 0.0371 | 0.0073 | 0.9012 | **0.9219** |
| corner | -0.0605 | -0.0892 | -0.1244 | -0.1221 | 0.4919 | **0.6079** |
| ca_easy | 0.1249 | 0.0151 | 0.0952 | -0.0019 | 0.8292 | **0.9743** |
| ca_hard | 0.0234 | 0.0021 | 0.0855 | -0.0123 | 0.8006 | **0.8581** |

Actor-Critic-based algorithm, MACPF, struggle to perform well on these tasks. Despite applying our adaptive TD($\lambda$) method to the QMIX algorithm, the observed performance improvement is marginal and falls short of matching the performance achieved by MAPPO.

The suboptimal performance of value-based MARL algorithms with mixing networks on Gfootball scenarios can be attributed to the intrinsic characteristics of the Gfootball environment. A prominent factor contributing to this is the sparse reward setting, where agents receive binary rewards solely upon scoring goals. Value-based approaches rely on accurate estimations of the value for each state-action pair through Temporal Difference (TD) updates, and the sparsity of rewards in this context amplifies the difficulty of this estimation process. In contrast, actor-critic algorithms delegate the role of selecting actions based on the largest distribution to actors, while critic networks are trained through discounted return calculations. Additionally, the Gfootball environment diverges from the Decentralized Partially Observable Markov Decision Process (Dec-POMDP) setting, as it lacks a global state. Most implementations concatenate observations and treat them as the state, a departure from the typical Dec-POMDP formulation. Analysis of the observation formation, according to the official documentation, reveals that observations encompass all available information and are not subject to partial masking. Moreover, in the checkpoint scoring settings of the Gfootball environment, rewards are provided for each agent individually instead of a global reward. This results in naturally separated rewards assigned to each agent. In QMIX-based algorithms, the mixing network is responsible for aggregating rewards, and the sparse reward setting intensifies the challenges associated with credit assignment processes.

## D MORE DISCUSSION ON SMACv2

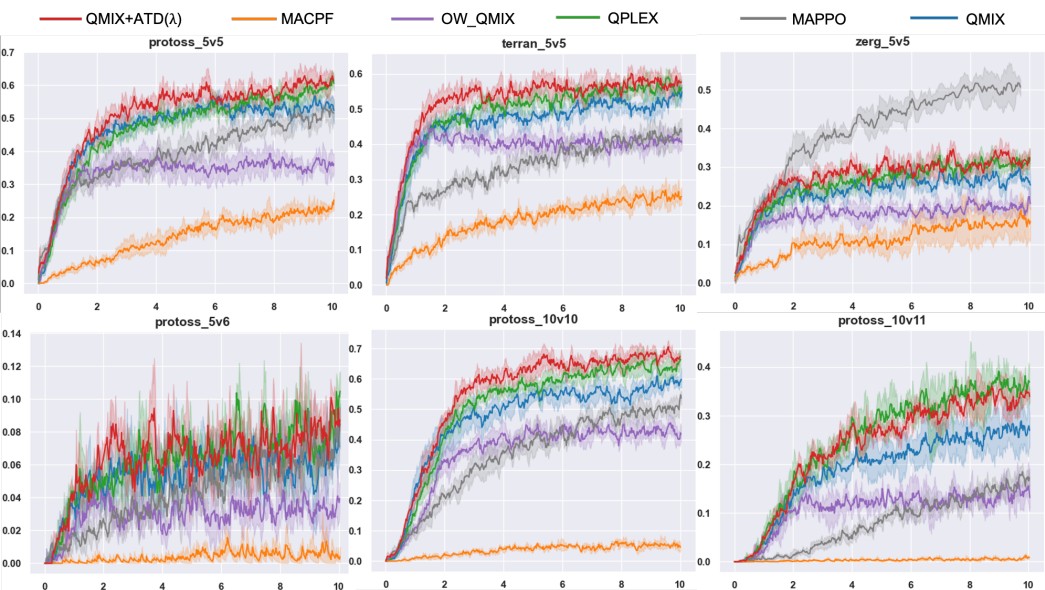

Figure 10: The winning rate curves evaluated on the 6 SMAC tasks with two major difficulties. The x-axis represents the time steps (1e6) being evaluated and the y-axis is the mean of the winning rate among 5 seeds.

We also verify our proposed adaptive $\lambda$ methods on 6 configs of SMACv2 (5 vs 5 of protoss, zerg, and terran), and (5 vs 6, 10 vs 10, and 10 vs 11 of protoss) with default unit generation probabilities. The unit generation policy is combined with 50% probability symmetric position and 50% probability of surrounding config. The difficulty is set as 7 by default. The winning rates of battles are calculated by the mean of 5 different seeds and smoothed by 0.8 for better visualization within 10M time steps.

The average test winning rate, computed across 5 seeds for each of the 6 scenarios, is depicted in Figure 10 to provide a comprehensive overview of the algorithms' overall performance. According to the figure, our proposed method shows marginal improvements in baseline algorithms. In the protoss_5v5, terran_5v5, and protoss_10v10 scenario, our method achieves slightly faster convergence

speed and higher performance. In the protoss_5v6 and protoss_10v11 scenario, our method can compete with the baseline algorithms. In the zerg_5v5 scenario, the MAPPO algorithm achieves the highest performance by a wide margin, but our method can also compete with other value-based baseline algorithms. In summary, in the SMACV2 environment with large randomness, our method does not improve the baseline algorithms significantly, however, our method also does not hold back the performance.

**Discussion and Limitation**     According to (Ellis et al., 2024), the SMACv1 environment lacks randomness in initializing the task settings. SMACv2 in contrast provides the configuration settings including the number of agents and enemies, the initialized positions, and the probability of generating a unit type. However, this configuration provides too much randomness, to some extent, instability. The large randomness makes duplicated transitions in the slow buffer difficult to occur, which makes the density ratio of that transition quite low. Therefore, the $\lambda$ values of the transitions quickly decay to 0, fully TD-update of Q functions, which is similar to other baseline algorithms. Meanwhile, due to the large variance taken by SMACv2 itself, the Monte-Carlo methods (TD(1)) will also provide a large variance which exacerbates the difficulty of convergence. Therefore, we believe that effective solutions to the SMACv2 problems still remain open in the future.

## E   MORE DISCUSSIONS ON SMAC TASKS

Table 2: Performance in SMAC tasks within 2M time steps

| Task | QMIX | QPLEX | OW-QMIX | MAPPO | MACPF | ATD |
|------|------|-------|---------|-------|-------|-----|
| 3m | 0.981 | 0.988 | 0.960 | 0.989 | **0.994** | 0.989 |
| 8m | 0.929 | 0.971 | 0.961 | 0.946 | 0.976 | **0.998** |
| 25m | 0.872 | 0.530 | 0.949 | 0.969 | 0.930 | **0.975** |
| 5m_vs_6m | 0.551 | 0.455 | 0.387 | 0.485 | 0.445 | **0.606** |
| 8m_vs_9m | 0.919 | 0.635 | 0.877 | 0.756 | 0.393 | **0.935** |
| 10m_vs_11m | 0.921 | 0.645 | 0.906 | 0.702 | 0.271 | **0.969** |
| 27m_vs_30m | 0.512 | 0.105 | 0.191 | 0.582 | **0.817** | 0.728 |
| MMM | 0.966 | 0.974 | 0.942 | 0.931 | **0.988** | 0.971 |
| MMM2 | 0.382 | 0.263 | 0.808 | 0.436 | 0.889 | **0.898** |
| 2s3z | 0.948 | 0.974 | 0.871 | 0.933 | **0.985** | 0.974 |
| 3s5z | 0.863 | 0.934 | 0.834 | 0.418 | **0.968** | 0.930 |
| 3s5z_vs_3s6z | 0.015 | 0.074 | 0.009 | 0.110 | 0.061 | **0.570** |
| 3s_vs_3z | 0.972 | **0.992** | 0.967 | 0.987 | 0.982 | 0.990 |
| 3s_vs_4z | 0.892 | 0.368 | 0.812 | **0.962** | 0.632 | 0.931 |
| 3s_vs_5z | 0.383 | 0.327 | 0.656 | **0.962** | 0.163 | 0.929 |
| 1c3s5z | 0.974 | 0.955 | 0.967 | **0.987** | 0.979 | 0.974 |
| 2m_vs_1z | 0.980 | 0.986 | 0.985 | 0.999 | 0.992 | **0.999** |
| corridor | 0.250 | 0 | 0 | 0.330 | 0.374 | **0.454** |
| 6h_vs_8z | 0.119 | 0.008 | 0.006 | 0.001 | 0.010 | **0.325** |
| 2s_vs_1sc | 0.982 | 0.991 | 0.932 | **0.999** | 0.992 | 0.984 |
| so_many_baneling | 0.926 | 0.953 | 0.925 | 0.967 | **0.981** | 0.961 |
| bane_vs_bane | 0.976 | 0.997 | 0.994 | 0.997 | 0.988 | **0.999** |
| 2c_vs_64zg | 0.922 | 0.823 | 0.901 | **0.954** | 0.945 | 0.931 |

**Performance on super-hard tasks.** In super-hard tasks, baseline algorithms and currently state-of-the-art algorithms hardly have acceptable results. In the 6h_vs_8z scenario, none of the algorithms mentioned in this paper converges to optimal policy within 2M time steps. In the 3s5z_vs_3s6z scenario, we carefully adjust the hyper-parameters as shown in Table 4, which provides larger exploration opportunities to agents to find a path towards winning results. Super-hard tasks make the Q value estimation more difficult and more exploration should be made before the policy improvements. Proper TD($\lambda$) value makes the Q value estimation of (s,a) pairs more accurate. Thus, the performance is much higher than 0 values or preset values.

According to Table 2, among 23 different tasks, our method achieves 11 best and 6 second best performances. The current state-of-the-art algorithm, MACPF, achieves 6 best and 7 second best performances. However, some of the easy scenarios cannot distinguish the performances among all

the scenarios. In the 5 super-hard subtasks including 2c_vs_64zg, 3s5z_vs_3s6z, MMM2, corridor, and 6h_vs_8z, our method achieves 4 best performances and MAPPO, as well as MACPF, achieve 1 best and 1 second best performances correspondingly, which indicates that our method can compete with and outperform sota AC-based and value-based baseline algorithms.

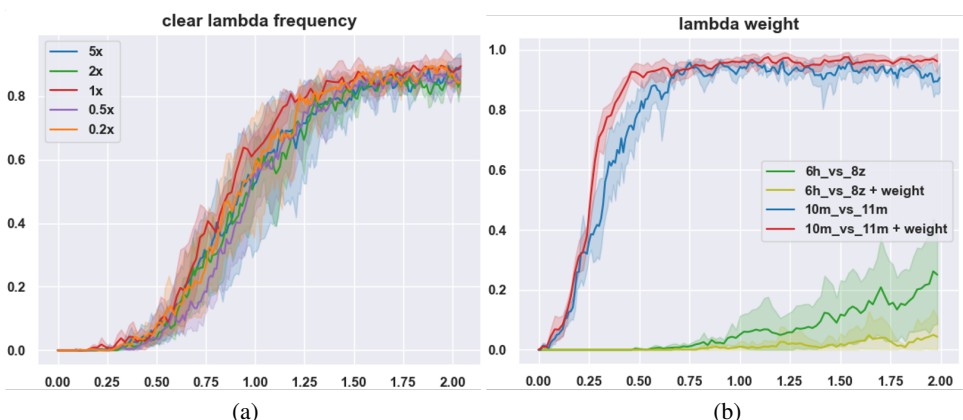

(a)    (b)

Figure 11: (a) The winning rate curves evaluated on MMM2 with different $\lambda$ cache frequency compared with $\theta^-$ update. (b) The winning rate curves of our method with/without importance weights evaluated on 10m_vs_11m (Hard), and 6h_vs_8z (Super-hard) scenarios. The x-axis is the time steps (1e6) and the y-axis is the average winning rate among 32 different seeds for 10 times experiment.

$\lambda$ **value cache**    To provide stable target value within the update interval of the lagged target network and update $\lambda$ values according to new policy distributions, we cache the calculated $\lambda$ values to the replay buffer and clear them with the target network update frequency. To test the influence of the cache frequency, we conduct an experiment on the MMM2 scenario and choose 5x, 2x, 1x, 0.5x, and 0.2x update frequencies of $\theta^-$.

As shown in Figure 11a, the 1x update frequency achieves the slightly highest performance and faster convergency speed. Compared with the 1x update frequency, lower update frequencies will result in the lag update of $\lambda$ values, which indicates that the $\lambda$ value cannot reflect the density ratio in time. In contrast, larger update frequencies might result in the instability of target values that the same transition may provide different target values given the same $\theta^-$ but different $\lambda$ values. Consequently, lower and larger clear frequencies also result in larger variances. Therefore, in this work, the cache frequency is recommended to be the same as that of target network parameters.

**The influence of importance weights.** In addition to the dynamic calculation of $\lambda$ value, the $\lambda$ value can be used as an importance weight onto the TD update after the self-normalization process (Sinha et al., 2022). Thus, to test the scope of the use of $\lambda$ weights, we apply the importance weight in the 10m_vs_11m (hard scenario) scenario and 6h_vs_8z (super hard scenario which needs much exploration).

According to Figure 11b, in the 10m_vs_11m scenario, applying the importance weight on the dynamic $\lambda$ method contributes to the convergence stability. In contrast, in the 6h_vs_8z scenario, the dynamic $\lambda$ without importance weights achieves higher performance. As described in (Rashid et al., 2020), the importance weight method makes the training process focus on the commonly appeared transitions. Therefore, for the hard tasks that do not need much exploration, importance weights help the utility network to reach more accurate values. On the opposite, for those super hard scenarios that need more exploration, the explored transitions that do not occur frequently are neglected. In conclusion, the importance weights can be applied to more exploitation scenarios and cannot be applied to exploration tasks. It is recommended when replayed trajectories diverge significantly from the current policy and it is unnecessary (and potentially harmful) when the policy is stable or slowly evolving.

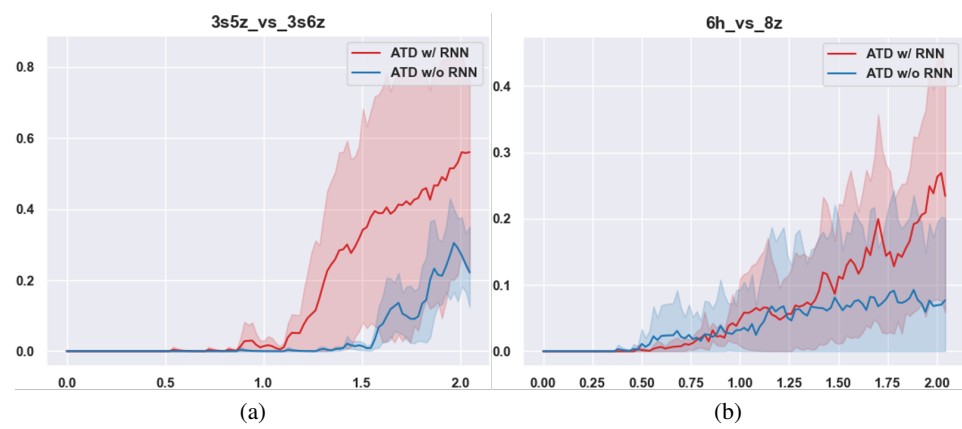

Figure 12: The winning rate curves of our method with/without RNN cells evaluated on 3s5z_vs_3s6z (Super Hard), and 6h_vs_8z (Super-hard) scenarios.

**The influence of RNN** In this paper, we set the network parameter of the ATD network similar to the utility network of each agent, so there is a GRU layer in the ATD network. To show the utility of the RNN layer, we test the performance of ATD+QMIX on 3s5z_vs_3s6z and 6h_vs_8z scenarios. According to the experimental results as shown in Figure 12, the ATD network with RNN layers achieves higher performance in early time steps. Empirically, the $\lambda$ values of the same transitions (s,a) in different trajectories should depend on the off-policy degree of the trajectories. In other words, the trajectories with different ages should possess different off-policy degrees. Based on this prerequisite, transitions within one trajectory should also depend on the density ratio. Therefore, (s,a) pairs with its history should contribute to the $\lambda$ value coordinately.

## F TRAINING DETAILS

### F.1 HYPER-PARAMETERS

Most of the hyper-parameters used in this paper are the default parameters from the codebase pymarl. The corresponding important parameters of SMAC and algorithms are listed below.

The QMIX algorithm we use is from pymarl code base (Samvelyan et al., 2019), the QPLEX and OW_QMIX are from pymarl2 code base (Hu et al., 2021), the MAPPO algorithm is from the official code-base (Yu et al., 2022) and MACPF is from the open-sourced code from paper (Wang et al., 2023). The detailed hyper-parameters are listed in Table 3 and the modified hyper-parameter for task 3s5z_vs_3s6z is shown in Table 4:

Apart from the hyper-parameters in the pymarl codebase. The hyper-parameters of the MAPPO algorithm are the default settings provided by the codebase. This codebase specifies corresponding hyper-parameters for each scenario. We change the total training time steps to 2M and the evaluation episodes to 6. When dealing with Gfootball tasks, the total training time step is 50M and rollout by 32 instances with the parallel runner.

### F.2 PSEUDO-CODE

## G BASELINE ALGORITHMS

**QMIX** QMIX is a value-based cooperative MARL algorithm that factorizes the joint action-value function into a monotonic mixing of individual agent utilities. Each agent learns a local utility network conditioned on its own observations, while a centralized mixing network, parameterized by hypernetworks and conditioned on the global state, combines these utilities into a joint value $Q_{\text{tot}}$. The monotonicity constraint $\partial Q_{\text{tot}}/\partial Q_a \geq 0$ ensures that maximizing $Q_{\text{tot}}$ can be achieved

Table 3: hyper-parameters for baseline algorithms

| parameter | QMIX+ATD | QMIX | QPLEX | OW_QMIX | MACPF |
|---|---|---|---|---|---|
| gamma | 0.99 | 0.99 | 0.99 | 0.99 | 0.99 |
| batch_size | 32 | 32 | 32 | 32 | 32 |
| buffer_size | 5000 | 5000 | 5000 | 5000 | 5000 |
| lr | 0.001 | 0.0005 | 0.0005 | 0.001 | 0.0005 |
| critic_lr | - | - | - | - | 0.0005 |
| optim_alpha | 0.99 | 0.99 | 0.99 | 0.99 | 0.99 |
| optim_eps | 0.00001 | 0.00001 | 0.00001 | 0.00001 | 0.00001 |
| rnn_hidden_dim | 64 | 64 | 64 | 64 | 64 |
| optim | RMSprop | RMSprop | RMSprop | RMSprop | RMSprop |
| action_selector | eps-greedy | eps-greedy | eps-greedy | eps-greedy | multinomial_seq |
| epsilon_start | 1.0 | 1.0 | 1.0 | 1.0 | 1.0 |
| epsilon_finish | 0.05 | 0.05 | 0.05 | 0.05 | 0.05 |
| epsilon_anneal_time | 50000 | 50000 | 50000 | 100000 | 50000 |
| agent_output_type | q | q | q | q | pi_logit |
| mixer | qmix | qmix | dmaq | qmix | dfop |
| mixing_embed_dim | 32 | 32 | 32 | 32 | 64 |
| hypernet_layers | 2 | 2 | - | 2 | - |
| hypernet_embed | 64 | 64 | 64 | 64 | 64 |
| adv_hypernet_layers | - | - | 3 | - | 1 |
| adv_hypernet_embed | - | - | 64 | - | 64 |
| td_lambda | 0.4 | 0.4 | 0.4 | 0.6 | 0.8 |
| double_q | False | False | True | True | False |
| num_kernel | - | - | 10 | - | - |
| is_minus_one | - | - | True | - | - |
| weighted_head | - | - | True | - | - |
| is_adv_attention | - | - | True | - | - |
| is_stop_gradient | - | - | True | - | - |
| central_mixing_embed_dim | - | - | - | 256 | - |
| central_action_embed | - | - | - | 1 | - |
| central_agent | - | - | - | central_rnn | - |
| central_rnn_hidden_dim | - | - | - | 64 | - |
| central_mixer | - | - | - | ff | - |
| n_head | - | - | - | - | 4 |
| attend_reg_coef | - | - | - | - | 0.001 |
| burn_in_period | - | - | - | - | 100 |
| dep_n_head | - | - | - | - | 4 |
| dep_embed_dim | - | - | - | - | 64 |
| dep_kv_dim | - | - | - | - | 64 |
| dep_output_dim | - | - | - | - | 64 |
| lfiw_optim | Adam | - | - | - | - |
| lfiw_optim_lr | 0.001 | - | - | - | - |

Table 4: Different hyper-parameters of 3s5z_vs_3s6z

| | |
|---|---|
| epsilon_start | 1.0 |
| epsilon_finish | 0.05 |
| epsilon_anneal_time | 100000 |
| batch_size | 128 |
| rnn_hidden_dim | 256 |
| hypernet_layers | 1 |
| hypernet_embed | 256 |
| optim | Adam |

via decentralized greedy actions. Its simplicity, scalability, and strong empirical performance make QMIX a widely used baseline for cooperative MARL.

---

**Algorithm 1** MARL with Adaptive TD($\lambda$)

---

1: Initialize action-value networks for all agents with parameters $\theta$, mixing network with parameter $\psi$, ATD network with parameter $\phi$, large replay buffer $D_{off}$ and small replay buffer $D_{on}$
2: Initialize target networks: $\psi' = \psi, \theta' = \theta$
3: **while** within the maximum number of time steps **do**
4:    set trajectory buffer $T = [\,]$
5:    **for** each environment step **do**
6:       collect new transition tuples $(s, a, r, s')$ with utility network $\theta$
7:       Store transition $(s, a, r, s')$ to $T$
8:    **end for**
9:    Store trajectory $T$ into $D_{on}$ and store overflowed trajectory from $D_{on}$ to $D_{off}$
10:    Sample a batch $B$ of training data from $D_{off}$ # Training utility and mixing network
11:    **for** each trajectory $T$ in $B$ **do**
12:       **for** each transition $(s, \tau, a, r, s', \tau')$ in $T$ **do**
13:          Compute $Q_i(\tau_i, a_i; \theta_i)$ for each agent $i$
14:          Compute $Q_{tot}(s, a; \psi)$
15:          Compute $\lambda$ value by $\omega_\phi(s, a)$
16:          Obtain target value for $\mathcal{R}^\pi Q'_\theta$ via $\lambda$
17:       **end for**
18:    **end for**
19:    Adam updates $\theta$ and $\psi$ with TD loss
20:    **if** Target Network Frequency **then**
21:       Update networks $\theta' = \theta$ and $\psi' = \psi$
22:       sample from $D_{off}$ and $D_{on}$
23:       update $\omega_\phi$ with loss function $L_\omega(\phi)$
24:    **end if**
25: **end while**

---

**QPLEX**   QPLEX extends value factorization by adopting an advantage-based decomposition known as individual global max (IGM) advantage learning. It leverages a duplex dueling architecture to decompose the global advantage into agent-wise advantages while respecting the IGM principle. This formulation relaxes the strict monotonicity constraints imposed by QMIX and allows the mixing network to represent more expressive coordination patterns. Consequently, QPLEX achieves improved performance in tasks with complex, non-monotonic inter-agent interactions.

**Weighted QMIX**   Weighted QMIX modifies the QMIX training objective by introducing adaptive importance weights on temporal-difference errors. This weighting mechanism mitigates overestimation bias and helps address credit assignment imbalance by adjusting the contribution of each training sample based on state importance or consistency among agent utilities. The algorithm retains the decentralized execution scheme of QMIX while offering enhanced stability, robustness, and sample efficiency through more informed value function updates.

**MAPPO**   MAPPO is a multi-agent adaptation of Proximal Policy Optimization designed under the centralized training with decentralized-execution (CTDE) paradigm. Each agent maintains an individual policy network conditioned on local observations, while a centralized critic value function utilizes privileged global state information during training to produce low-variance advantage estimates. MAPPO applies clipped surrogate objectives, generalized advantage estimation (GAE), entropy regularization, and value normalization to stabilize learning across multiple agents. Owing to its robustness and strong empirical results, MAPPO has become a dominant policy-based baseline for cooperative MARL.

**MACPF**   MACPF (Multi-Agent Conditional Policy Factorization) introduces a conditional factorization of the joint policy to enhance multi-agent coordination while preserving decentralized execution. The algorithm represents the joint policy as a chain of conditional individual policies, $\pi(a) = \prod_{i=1}^n \pi_i(a_i \mid o_i, a_{<i})$, allowing each agent to condition its action not only on its observation but also on the actions of preceding agents in an ordered factorization. During centralized training, this structure enables the actor and critic to capture rich inter-agent dependencies and more accurate

advantage estimates over joint actions. During execution, agents act in a decentralized manner by following the predetermined factorization order and conditioning only on information available through this ordering. This design yields more expressive coordination than independent policies while remaining fully compatible with the CTDE framework.

## H    WALLTIME AND HARDWARE FOR TRAINING

We operate our experiments on servers with 3.9 python version, AMD EPYC 7543 32-Core Processor CPU, and NVIDIA GeForce RTX 3090 GPU. The maximum interaction time steps is 2.05M including test episodes and the StarCraftII version is 4.10. We set up 5 experiments with different seeds simultaneously and the actual time spent is about 6 hours and 30 minutes per task. For SMACv2, it takes rougly 20 hours to finish 10M training time steps with parallel runner and more units will increase the rollout time. For the Gfootball environment, we set 32 rollout threads and 50M time steps. The actual average time spent is about 29 hours for MAPPO, 35 hours for our method, 47 hours for QMIX, and 90 hours for MACPF.

