# OpenReview forum: "Adaptive TD-Lambda for Cooperative Multi-agent Reinforcement Learning"
_ICLR.cc/2026/Conference — ICLR 2026 Conference Withdrawn Submission_

### Official Review · Reviewer_MhPt · 2025-10-16

**Soundness:** 1
**Presentation:** 1
**Contribution:** 2
**Rating:** 2
**Confidence:** 4

**Summary:**

This paper proposed ATD($\lambda$), a novel method to automatically set $\lambda$ for TD learning. To do so an estimation network is used, which is trained to compute the $f$-divergence between an off-policy bigger dataset and an on-policy smaller one. The proposed method can be plugged in every method doing TD learning, both value-based algorithms and for critics in actor-critic methods.

**Strengths:**

TD($\lambda$) is a very popular algorithm for TD learning, and usually the setting of $\lambda$ is both a delicate and sensible choice. Thus, being able to automatically determine such value is a valid contribution to the field. Also, I like the idea of using two different datasets to determine such value, to express the degree of off-policy-ness of a transition. Experimental results seem quite good, and provide a valid support to the claimed benefits of the method.

**Weaknesses:**

The paper itself does not look great: the explanations are often quite convolved and difficult to follow, mixing mathematical details with implementation ones. The space given to the exposition of your own proposed method is very short. Also, in terms of experimental results, quite often you are mentioning results that are not included in the main paper, and this is not a good practice.

**Questions:**

- The paragraph explaining the $\lambda$-return extension in the Background is quite difficult to actually understand. I would suggest to rephrase it to be a lot clearer and more detailed.

- Most of Section 4.1 does not seem very useful: QMIX has already been introduced in the Background, and not being a contribution of yours, I do not see why you employ an entire page to get into so many details (including implementation ones, like config files, which are not extremely useful in a ML paper at this stage) for something that is not really your work...

- It is not very clear how Equation (8) follows from (7) when we get $f$ to be the Jensen-Shannon divergence. Perhaps, given that this is one of the focal point of the paper, it would deserve a more detailed explanation.

- You mention actor-critic methods quite a lot in the introduction of the paper, and claim that you are combining your ATD($\lambda$) with MAPPO. However, I do not see any explanation on how this integration is done. For example, you say that your method requires minimal changes (i.e., only to add the on-policy small buffer), but this holds true for value-based methods (which are off-policy), but not for actor-critic algorothms like MAPPO. For those for example you need to add both buffers, resulting in quite some additional resources usage.

- Figure 4 is partly covering the caption of Figure 3. You probably modified some margins to get them to fit, but this should not be done...

- $Q$-learning based methods like QMIX are off-policy, but they do not require importance sampling. Why did you decide to integrate it as one of the baselines? How is it used in there?

- In Figure 5 caption, you say that MMM2 is a super-hard map, but before you said it is only hard. Which is the correct one?

- Actually, I do not see any result for MAPPO in the paper... Then why do you claim you have extended your ATD($\lambda$) to this as well?

- The same goes for results on GRF: there is none in the paper.

---

> ### Author Response · Authors · 2025-11-27
> **Respond to Weaknesses and Questions (1/2)**
>
> We thank the reviewer for the detailed comments and the opportunity to improve the clarity and presentation of the paper. Below, we respond to all concerns. We now try our best to respond to each point in detail below.
>
> > W1: The paper itself does not look great: the explanations are often quite convolved and difficult to follow, mixing mathematical details with implementation ones. The space given to the exposition of your own proposed method is very short. Also, in terms of experimental results, quite often you are mentioning results that are not included in the main paper, and this is not a good practice.
>
> We appreciate the reviewer’s feedback. In the revised version, we will significantly improve the clarity and structure of the text. We will restructure Sections 3 and 4 so that the theoretical derivation of ATD($\lambda$) is presented cleanly and self-contained. We also substantially lengthen Section 3, adding clearer derivations, intuitive explanations, and algorithm summaries. Due to the page limit, we try our best to show more content in the main paper and have to link some of the results to the Appendix. We believe these changes will improve readability and organization.
>
> > Q1: The paragraph explaining the $\lambda$-return extension in the Background is quite difficult to actually understand. I would suggest to rephrase it to be a lot clearer and more detailed.
>
> Thank you for pointing this out. We agree that the paragraph was overly compressed. In the revision, we rewrite the lambda return part. The modified content is shown in color violet.
>
> > Q2: Most of Section 4.1 does not seem very useful: QMIX has already been introduced in the Background, and not being a contribution of yours, I do not see why you employ an entire page to get into so many details (including implementation ones, like config files, which are not extremely useful in a ML paper at this stage) for something that is not really your work..
>
> We partially agree with the reviewer to introduce the QMIX algorithm. The detailed QMIX exposition was included to ensure reproducibility and guarantee the logic flow of our proposed ATD method. Meanwhile, we also want to show the compatibility of our ATD and take the QMIX architecture as an example. Some latter reviewers asked about the mention of QMIX such that the readers who are not quite familiar with the QMIX can follow our architecture. Based on that, we modify the overall architecture and separate the contents by QMIX paragraph and ATD paragraph in this revised version.
>
> > Q3: It is not very clear how Equation (8) follows from (7) when we get $f$ to be the Jensen-Shannon divergence. Perhaps, given that this is one of the focal point of the paper, it would deserve a more detailed explanation.
>
> Sorry for the confusion. We have added the section on mathematical connecting Jensen-Shannon divergence and $f$ with BCE loss in Appendix A.4. In this current revised version, due to the space limit, we add the reference to the A.4 section.

---

> ### Author Response · Authors · 2025-11-27
> **Respond to Questions(2/2)**
>
> > Q4: You mention actor-critic methods quite a lot in the introduction of the paper, and claim that you are combining your ATD($\lambda$) with MAPPO. However, I do not see any explanation on how this integration is done. For example, you say that your method requires minimal changes (i.e., only to add the on-policy small buffer), but this holds true for value-based methods (which are off-policy), but not for actor-critic algorothms like MAPPO. For those for example you need to add both buffers, resulting in quite some additional resources usage.
>
> Thank you for pointing out this confusion. We intended to explain that ATD($\lambda$) is compatible with value-based training and the critic network of  MAPPO is value-based, but the explanation was not clear. For MAPPO, we maintain two buffers. The on-policy buffer is the standard on-policy rollout buffer used by MAPPO, the size of which should be the rollout threads defined in MAPPO. The original on-policy buffer is the trajectory buffer. Based on that, we add an off-policy buffer with a size of 50x. In the original MAPPO, the actor is trained via GAE from the on-policy buffer and the critic is trained by the MSE loss between the critic value and the expected cumulated return. We change the cumulated return (MC return) into the ATD calculation with the data from both on/off-policy buffers. In such cases, the critic training process is slightly changed and the off-policy buffer is added. We add this in the training pipeline section (4.3). As for the ATD+MAPPO results, we conduct experiments on Google Football academic scenarios, also due to the space limit, which is shown in Appendix C in this revised version.
>
> > Q5: Figure 4 is partly covering the caption of Figure 3. You probably modified some margins to get them to fit, but this should not be done...
>
> We apologize for this formatting issue, which arose during final compilation. We correct the margin and ensure all figures are separated properly, which should be normal in the current revised version.
>
> > Q6: $Q$-learning based methods like QMIX are off-policy, but they do not require importance sampling. Why did you decide to integrate it as one of the baselines? How is it used in there?
>
> We include QMIX as a baseline not only because it closely matches our algorithmic structure, but also because QMIX is one of the standard and strongest value-based MARL baselines in SMAC. As described in RIIT[1], QMIX with enough implementation tricks can achieve high performance, and one of the tricks is the $\lambda$ value choice. Meanwhile, a common choice of $\lambda$ value selection, such as Vtrace and Retrace, is based on the importance sampling mechanism. QMIX itself does not require importance sampling and we are not combining the QMIX with importance sampling, but calculate the $\lambda$ value via importance sampling and apply that $\lambda$ value to TD learning for training QMIX. We rely on standard replay buffer sampling as in the original QMIX implementation.
>
> > Q7: In Figure 5 caption, you say that MMM2 is a super-hard map, but before you said it is only hard. Which is the correct one?
>
> Thank you for noticing this inconsistency. The correct classification in the SMAC benchmark is “Super Hard.” We will ensure consistent terminology throughout the paper.
>
> > Q8: Actually, I do not see any result for MAPPO in the paper... Then why do you claim you have extended your ATD($\lambda$) to this as well?
> >
> > Q9: The same goes for results on GRF: there is none in the paper.
>
> Sorry for the misleading content. I applied the ATD method to MAPPO and conducted experiments on the GRF environment, which was not in the main paper in the previous version. In this current revised version, I have added the MAPPO adaptation in the Method section and put the GRF experiment results in Appendix C.
>
> [1] Rethinking the Implementation Tricks and Monotonicity Constraint in Cooperative Multi-Agent Reinforcement Learning
>
>
>
> We thank the reviewer again for the helpful feedback and will incorporate the improvements into the final manuscript. We believe that addressing these points will provide a clearer understanding of our work and its contribution to the field. We are more than willing to provide any further information or clarification that might be required for a more comprehensive evaluation.

---

> > ### Comment · Reviewer_MhPt · 2025-11-27
> > **Reply to the Authors**
> >
> > I would like to thank the authors for their rebuttal, which addressed some of my concerns. Here are some follow-up points:
> >
> > **Q2:** While it is absolutely fair (and useful indeed!) to introduce QMIX for the unfamiliar readers, this already happened in the Background section (which seems like the natural place for that). What I was arguing is that there is no need to provide such a detailed explanation, like the one in Section 4.1, of a method that you are not proposing yourself. This takes a lot of space, space that you could have used to better present your own idea or some additional results, and in general it does not make too clear that QMIX is no contribution of yours, but rather a starting point from the literature that you are building upon. Even the very precise implementation details are given, which, other than not being useful here, are also very restricted to the specific MARL framework you used (pymarl, I suppose), making these not even generalizable to other different frameworks (like BenchMARL for example), probably only confusing the readers which are used to different frameworks rather than helping them.
> >
> > **Q4:** Then I am right in saying that the integration of your proposed ATD($\lambda$) does not ALWAYS come almost for free: if you are integrating it into on-policy algorithms like MAPPO, which thus do not have a large off-policy replay buffer, this requires a substantial amount of additional memory to be maintained and used.
> >
> > **Q6:** I was not arguing on why you included QMIX as a baseline (it seems an obvious choice indeed: it is a one of the most popular MARL algorithms out there, and is one of the methods you are building upon), but rather on how you integrated IS into a Q-learning-based algorithm (which is known for being off-policy but not require IS, as it uses the greedy policy as its behavioural policy). I see you are using IS to determine $\lambda$ rather than adjust the updates themselves, but the connection between IS and the $\lambda$ parameters are not entirely clear to me from the paper: has such connection been shown somewhere? What is the intuition behind it? How is this specific integration done? I see that both your contribution here and popular methods like Vtrace or Retrace builds on this intuition, but I think that a clearer explanation of how this is integrated into QMIX and why would be extremely helpful for the reader.

---

> > > ### Author Response · Authors · 2025-12-04
> > > **Reply to some follow up concerns**
> > >
> > > We would like to thank the reviewer for the detailed follow-up comments. We are trying our best our respond the the continued concerns.
> > >
> > > **About Q2**: Thank you for elaborating on this point, and we fully agree with this point. We acknowledge that we describe the QMIX algorithm in the current Section 4.1, which has already been introduced in the Background section and is actually not our contribution. Therefore, we decided to add a section in the appendix and describe the baseline algorithms. Meanwhile, we move the 'influence of RNN' paragraph to Section E (More discussion on SMAC) of the Appendix. The space reserved is used to show the empirical influence of $\lambda$ value assignment on the credit assignment process. The changes are shown in the latest revision.
> > >
> > > **About Q4**: Yes, we agree to the reviewer's interpretation, and we appreciate the chance to clarify. For value-based off-policy methods, integrating ATD is almost free because they have already stored the full trajectory in large buffers and the small buffer is 50 times smaller, which slightly increases the memory cost. In contrast, for on-policy actor-critic methods, we have to add a 50 times larger buffer to store historical data, which is a large amount of memory cost. In the previous version of the paper, we wanted to claim that the ATD integration is 'technically simple' rather than memory cost-free. However, the additional memory cost is important, and we clarify that in the current revised version.
> > >
> > > **About Q6**: About the connection between $\lambda$ and the IS value, we believe that the paper '*Munos et al., “Safe and Efficient Off-Policy Reinforcement Learning” (2016)*' might have responded to this question. In this paper, the $c_t$ weight in the return-based operator can be replaced by IS value, $\lambda$ value from $Q^\pi $ algorithm, $\lambda\pi(a_s|x_s)$ from TB($\lambda$), and $\lambda min(1, IS)$ from Retrace. This mechanism adjusts the effective trace length according to how closely the behavior and target policies match: when they are similar, $c_t$ remains large and multi-step propagation is encouraged; when they diverge, $c_t$ shrinks and the return becomes more one-step, yielding a stable and safe target. We adopt the same idea: QMIX’s learning rule remains unchanged, and the adaptive $\lambda$-return simply replaces the usual one-step target.
> > >
> > > Another point is that QMIX uses an $\epsilon$-greedy behavior policy does not conflict with this mechanism. ATD was explicitly designed to handle arbitrary policies, including deterministic greedy policies. In such greedy or near-greedy cases, the IS ratio acts only as a diagnostic measure of behavioral mismatch: if the target policy selects the same greedy action, the IS value is $1$ and longer traces are safe; if it selects a different action, the IS value becomes small or $0$, automatically shortening the trace. However, in the MARL setting, the actual IS value is the multiplication of the IS value of each agent, which should result in a near-zero value. Therefore, similar to policy-based methods, we take the softmax value of $Q(\tau,a)$ functions as the policy and conduct comparison experiments based on that.

---

### Official Review · Reviewer_nB5g · 2025-10-21

**Soundness:** 3
**Presentation:** 3
**Contribution:** 2
**Rating:** 4
**Confidence:** 4

**Summary:**

The paper proposes ATD($\lambda$): a classifier-based, likelihood-free scheme that learns a per-transition weight $\omega$ from a pair of replay buffers
(recent on-policy vs older off-policy), maps $\omega$ via a sigmoid to a per-transition $\lambda$, and plugs this adaptive trace into TD($\lambda$) targets for centralized critics in MARL (QMIX, MAPPO).
Experiments on SMAC and GRF show empirical improvements over several fixed-$\lambda$ baselines and some off-policy correction methods.

**Strengths:**

- Clear and practical motivation: tuning $\lambda$ in large, non-stationary multi-agent settings is hard; an automatic per-transition scheme is appealing.

- The method is modular and easy to insert into existing CTDE pipelines (QMIX, MAPPO), which increases engineering utility.

- Empirical evaluation covers multiple MARL benchmarks and includes several ablations and baselines.

- The authors include derivations and attempt to provide theoretical justification for contraction and stability under certain assumptions.

**Weaknesses:**

1. The core mechanism — a classifier to distinguish recent vs older replay samples, using that output as a per-transition weight and mapping it to $\lambda$ — is not inherently multi-agent.
It can be applied identically in single-agent RL.
The paper’s contributions are largely (a) the idea of learned per-transition $\lambda$ via a two-buffer classifier and (b) engineering how to integrate it into CTDE critics.
There are no algorithmic changes that exploit MARL-specific structure.
This may lead readers to suspect that MARL was chosen primarily because its baselines are easier to improve rather than because the problem requires a multi-agent treatment.

2. The paper treats $\omega$ (sigmoid of classifier output) as the adaptive $\lambda$ and argues intuitively that on-policy samples should get larger $\lambda$ and off-policy smaller $\lambda$.
However, there is no formal definition of an optimal $\lambda$ (e.g., minimizing TD error MSE, minimizing update variance, maximizing sample efficiency) and no proof that the classifier-derived $\lambda$ approximates such an optimum.
The classifier learns a surrogate “on-policyness” score, not the $\lambda$ that would minimize any well specified loss.

3. The paper emphasizes that the method relates to off-policy correction (importance sampling), but the equivalence or relationship between the proposed objective and classical importance weighting is not clearly established.
It remains unclear whether the resulting target is unbiased or systematically biased by the sigmoid mapping and classifier errors.

4. Classifier-based density ratio estimation is known to be high-variance or biased in regimes with high overlap or low support.
The paper does not provide implementation details (update frequency, batch balancing, network capacity, regularization, etc.) nor quantitative diagnostics showing how classifier quality affects critic performance.
The observation that $\lambda$ collapses to 0 in highly stochastic settings (e.g., SMACv2) is mentioned but not analyzed in depth.

**Questions:**

1. Can the authors theoretically or empirically demonstrate that the learned λ leads to a bias–variance tradeoff closer to an optimal value (e.g., minimizing TD error MSE)?

2. Please include a clear pseudocode or algorithm box describing the full training loop, including how the two replay buffers are maintained, how the classifier is trained, and how λ is cached or updated.

3. A simple λ-annealing baseline (e.g., linearly or exponentially decaying λ as in [1] could serve as a stronger and more computation-efficient comparison.

[1] Revisiting Cooperative Off-Policy Multi-Agent Reinforcement Learning. ICML 2025.

---

> ### Author Response · Authors · 2025-11-27
> **Respond to Weaknesses (1/2)**
>
> We thank the reviewer for the constructive comments and valuable suggestions. We now try our best to respond to each point in detail below.
>
> > W1: The core mechanism — a classifier to distinguish recent vs older replay samples, using that output as a per-transition weight and mapping it to $\lambda$— is not inherently multi-agent. It can be applied identically in single-agent RL. The paper’s contributions are largely (a) the idea of learned per-transition $\lambda$ via a two-buffer classifier and (b) engineering how to integrate it into CTDE critics. There are no algorithmic changes that exploit MARL-specific structure. This may lead readers to suspect that MARL was chosen primarily because its baselines are easier to improve rather than because the problem requires a multi-agent treatment.
>
> We agree that the learned adaptive $\lambda$ mechanism could, in principle, be applied to single-agent RL algorithms. However, applying it in MARL introduces challenges that are not present in the single-agent setting, such as the difficulty of applying importance sampling to joint policies when the number of agents is large. Meanwhile, the replay data is generated by a non-stationary sequence of multi-agent joint policies, which drastically amplifies policy–data mismatch and makes fixed-$\lambda$ TD particularly unstable. Another point is that the state–action visitation overlap is harder to ensure in MARL. In many environments, a joint state-action tuple may never recur again, making sample-based estimation of density ratios more challenging. We believe that MARL faces these challenges that SARL does not, and these challenges motivate the design of our proposed likelihood-free ratio estimator that does not require access to explicit joint policies, and a two-buffer system that approximates on-policy vs. historical state-action distributions, enabling per-transition $\lambda$ estimation at scale.
>
> > W2: The paper treats $\omega$ (sigmoid of classifier output) as the adaptive $\lambda$ and argues intuitively that on-policy samples should get larger $\lambda$ and off-policy smaller $\lambda$ . However, there is no formal definition of an optimal $\lambda$ (e.g., minimizing TD error MSE, minimizing update variance, maximizing sample efficiency) and no proof that the classifier-derived $\lambda$ approximates such an optimum. The classifier learns a surrogate “on-policyness” score, not the $\lambda$ that would minimize any well specified loss.
>
> We appreciate the reviewer’s concern and clarify the rationale. The theoretical foundation we follow is based on the classical understanding of $\lambda$​ as controlling the bias–variance tradeoff of TD learning: Higher $\lambda$ approaches Monte Carlo estimation, lowering bias but increasing variance, and lower $\lambda$ approaches TD(0), yielding low variance but high bias under policy mismatch. In off-policy learning: $\hat{V}^{MC}=E*{d^\mu}[G_t],$ and $\hat{V}^{TD(0)} = \mathbb{E}*{d^\pi}[r + \gamma V(s')]$ ($E$ is the expectation. I have no idea why using '\mathbb' will cause error display).  MC estimates are unbiased only when sample trajectories match the current policy distribution $d^\pi$. Thus, the optimal $\lambda$ should increase when $d^D(s,a) \approx d^\pi(s,a)$ and decrease when the mismatch is larger. This motivates the connection $\lambda(s,a) \propto \text{similarity}(d^D, d^\pi)$. In single-agent theory [1,2], such a $\lambda$ can be formally shown to minimize the MSE of TD estimates under off-policy learning. In MARL, deriving a closed-form optimal $\lambda$ is currently intractable due to the lack of explicit policy distributions, which is precisely why we instead learn a classifier-based proxy for on-policyness, and use it to approximate $\lambda$ automatically and adaptively. In experiments, using this learned $\lambda$ reduces TD target error and improves convergence speed (shown in Figs. 3–6).

---

> ### Author Response · Authors · 2025-11-27
> **Respond to Weaknesses (2/2)**
>
> > W3: The paper emphasizes that the method relates to off-policy correction (importance sampling), but the equivalence or relationship between the proposed objective and classical importance weighting is not clearly established. It remains unclear whether the resulting target is unbiased or systematically biased by the sigmoid mapping and classifier errors.
>
> Thank you for pointing this out. In classical off-policy correction, $\omega(s,a) = \frac{\pi(a|s)}{\mu(a|s)}$ corrects the MC return to remain unbiased. However, in MARL $\pi$ and $\mu$ are not directly obtainable in joint form, importance ratios become intractable and high-variance, and the joint policies change rapidly during learning. Our approach instead learns a likelihood ratio proxy $\omega(s,a) \approx \frac{d^\pi(s,a)}{d^D(s,a)}$ without requiring explicit access to $\pi$ and $\mu$ and uses this quantity not to reweight returns directly, but to modulate $\lambda$ in the TD target $R_t = r_t + \gamma\big[\omega R_{t+1} + (1-\omega)\max Q(s',a')\big]$. Thus, when the transition is on-policy ($\omega$ large), MC returns are trusted more and when off-policy, TD bootstrap dominates. Therefore, our targets remain controlled-bias estimators rather than unbiased ones. They intentionally introduce some bias for stability and sample efficiency, which is similar to methods such as Retrace, V-trace, or COP-TD, but without requiring explicit IS ratios.
>
> > W4: Classifier-based density ratio estimation is known to be high-variance or biased in regimes with high overlap or low support. The paper does not provide implementation details (update frequency, batch balancing, network capacity, regularization, etc.) nor quantitative diagnostics showing how classifier quality affects critic performance. The observation that $\lambda$ collapses to 0 in highly stochastic settings (e.g., SMACv2) is mentioned but not analyzed in depth.
> >
> > Q2: Please include a clear pseudocode or algorithm box describing the full training loop, including how the two replay buffers are maintained, how the classifier is trained, and how $\lambda$ is cached or updated.
>
> We acknowledge this feedback and will include more details in the current revised version. We mention some of the details in the architecture section and emphasize them in blue. Meanwhile, we add a pseudo-code section in Appendix F.2, which might improve the clarity of our proposed method. Regarding the collapse in SMACv2, we add a section called 'More discussion SMACv2' in Appendix D and analyze the reason for $\lambda$'s collapsing to 0 in orange color.
>
> | ENV           | QMIX  | QPLEX     | WQMIX | MAPPO     | MACPF | ATD+QMIX  |
> | ------------- | ----- | --------- | ----- | --------- | ----- | --------- |
> | terran_5v5    | 53.41 | 56.16     | 41.01 | 42.28     | 25.18 | **57.32** |
> | zerg_5v5      | 27.14 | 30.96     | 19.75 | **50.39** | 16.05 | 32.06     |
> | protoss_5v5   | 53.93 | 59.34     | 35.95 | 50.85     | 22.47 | **60.77** |
> | protoss_5v6   | 6.51  | 7.88      | 3.23  | 6.93      | 0.59  | **8.33**  |
> | protoss_10v10 | 58.68 | 64.06     | 42.81 | 50.69     | 5.25  | **67.47** |
> | protoss_10v11 | 27.37 | **35.99** | 14.25 | 16.28     | 0.65  | 33.34     |
>
> The summarization of the newly added is: We evaluate our ATD methods on six SMACv2 configurations with default settings and compute performance across 10 million time steps averaged over five seeds. Results show that our method brings small improvements, slightly faster convergence and higher performance in several scenarios. However, our method generally performs similarly to baseline algorithms and does not significantly improve or harm performance overall. The performance overview demonstrates that the adaptive $\lambda$ method competes well with value-based baselines, though in some cases, such as Zerg 5v5, other algorithms like MAPPO perform better. In the discussion, we claim that SMACv2 introduces substantial randomness in unit types, counts, and positions. This high variance reduces repeated transitions between the slow and fast buffer, causes $\lambda$ values to decay quickly, and leads to updates similar to standard TD learning. As a result, in SMACv2, our ATD method performs at least **no worse** than baseline algorithms.

---

> ### Author Response · Authors · 2025-11-27
> **Respond to Questions**
>
> > Q1: Can the authors theoretically or empirically demonstrate that the learned $\lambda$ leads to a bias–variance tradeoff closer to an optimal value (e.g., minimizing TD error MSE)?
>
> Thank you for this question. While deriving a closed-form expression for the optimal $\lambda$ that minimizes TD-target MSE is intractable in large-scale multi-agent settings with function approximation and non-stationary joint policies, our method is grounded in the classical theoretical understanding of bias–variance tradeoffs in TD learning. Specifically, when a transition comes from a distribution close to the current policy $d^{\pi}$, Monte-Carlo style targets incur lower bias, whereas samples that are far off-policy accumulate high variance if used with MC returns and therefore benefit from stronger bootstrapping.
>
> For example, intuitively, suppose that a utility network improves and converges to 8 (10 is optimal) during the training process, the return of the historical trajectories is increasing to 10 from 1. Based on the off-policy training, the training data may be randomly sampled from the buffer. If the oldest trajectories are sampled, the MC return is 1 and the TD target is 10, the proper update method is fully TD(0). In contrast, suppose that the latest trajectory is sampled, the TD target is 8 and the MC return is 10, of which the MC should be used. Empirically, our results demonstrate that the learned $\lambda$ indeed leads to more accurate and more stable value learning. As shown in our experiments, the adaptive method reduces TD-target prediction error early in training, converges faster, and yields higher final performance across SMAC and GFootball tasks compared to any fixed-$\lambda$ baseline. The fact that performance improves even relative to strong off-policy baselines such as Retrace and static $\lambda$ schedules indicates that the learned $\lambda$ is better capturing the point along the bias–variance continuum that improves sample efficiency and value estimation quality. Although providing a formal MSE-optimal proof is unrealistic in large-scale MARL with deep function approximation, the combination of theoretical grounding, stable contraction properties, and clear empirical gains strongly supports that the learned $\lambda$ drives the method toward a superior bias–variance operating point in practice.
>
> > Q3: A simple λ-annealing baseline (e.g., linearly or exponentially decaying λ as in [3] could serve as a stronger and more computation-efficient comparison.
>
> We appreciate this suggestion. We have carefully read this research paper and found the $\lambda$-annealing method written in the paper. However, this paper currently does not provide the official code link and we find multiple implementation contributions in this paper. Meanwhile, the hyper parameters used in this paper are different from ours, including the parallel runner and the episode runner we use, and the batch size for training. As a consequence, the proper $\lambda$ value is different based on these two different replay ratios. As described in our manuscript, if the algorithm uses a parallel runner during the training process, the $d^\pi$ should be more similar to $d^D$ because the large buffer contains fewer policy trajectories. Therefore, the optimal $\lambda^\star$ provided in [3] should be different from our methods. However, an optimal $\lambda^\star$ introduced in [3] is another important hyper-parameter that affects the final performance. We have updated the result and replaced the $\lambda=1$ learning curve with 'annealing $\lambda$' in Figure 4. The annealing method is used according to [3]:
> $$
> \lambda_k=\lambda^\star+\frac{1-\lambda^\star}{1+\alpha k}
> $$
> Experimental results show that only the $\lambda$ annealing mechanism is not comparable to static value settings; it should be combined with other mechanisms described in [3]. Despite the difference in implementation, [3] should be an inspiration for our work and improve the reliability of our work. We have added this to our related work.
>
> [1] Doubly Robust Off-policy Value Evaluation for Reinforcement Learning
>
> [2] Error Bounds for Approximate Policy Iteration
>
> [3] Revisiting Cooperative Off-Policy Multi-Agent Reinforcement Learning. ICML 2025.
>
>
>
> We thank the reviewer again for the helpful feedback and will incorporate the improvements into the final manuscript. We believe that addressing these points will provide a clearer understanding of our work and its contribution to the field. We are more than willing to provide any further information or clarification that might be required for a more comprehensive evaluation.

---

### Official Review · Reviewer_R6oj · 2025-10-26

**Soundness:** 2
**Presentation:** 2
**Contribution:** 1
**Rating:** 2
**Confidence:** 4

**Summary:**

The authors propose ATD (Adaptive TD Lambda) which uses a smaller on-policy buffer to compute the off-policyness of the sampled trajectories. ATD learns a separate network $\omega_\phi$ which estimates the density ratio with a lower bound objective. Experimental results are provided for SMACv1 and GRF.

**Strengths:**

The approach is relatively simple, and adjusting $\lambda$ automatically with the density ratio computed from the two replay buffers is a nice idea. However, as I mention below in Weaknesses, the authors need to clarify the line between ideas/contributions from previous work and ATD.

**Weaknesses:**

1. The contributions of this work are not clear. There are some phrases like “inspired by” Sinsha et, al. 2022, Grover et, al. 2019), or Hu et, al. 2021. but it is not clear from the text what the main contribution is or what the challenge is of applying these techniques to MARL.
1. Related to the first point, if the off-policyness can be computed over states and joint actions, then it is a simple application of previous techniques to MARL. There is nothing specific to MARL which makes it hard to apply these techniques, such as factorized policies or the exponential complexity of the joint state/action space.
1. The experiments are limited to saturated benchmarks such as SMAC and GRF. Although the authors acknowledge that it cannot be applied well to SMACv2, that is a significant limitation as it is recently a more standard benchmark than SMACv1. The reason for this limitation is not provided in Appendix C (as they mention in the Conclusion). I suggest the authors go deeper into why this limitation persists, as it might provide hints for some challenges in MARL, and later some additional modifications to the current approaches which will be more novel in the MARL context.
1. In the background section, the “true state of the environment” in line 132 is not related to centralized training. The Dec-POMDP is a formulation, and centralized training is a choice on how to train policies. Similar comment holds for line 137 where partial observability is a characteristic of the formulation/setup of the environment, and it is not related to decentralized execution. Please separate these  in the text as it will confuse readers.
1. The observations $z$ for line 136 should have agent Id conditioning.
1. The notation $T$ is used twice to indicate the observation-action history and the target value in Eq.1.

**Questions:**

1. What is the connection between off-policyness and $\lambda$? What is the justification behind Eq. 9-10? My understanding is that if the density ratio $\omega$ is high, we have higher off-policyness and thus we would want to go more towards TD(0), but it seems like the reverse is happening in Eq. 9-10.
1. Please address my comments in Weaknesses.

---

> ### Author Response · Authors · 2025-11-27
> **Respond to Weaknesses (1/2)**
>
> We thank the reviewer for the constructive comments and valuable suggestions. We now try our best to respond to each point in detail below.
>
> > W1: The contributions of this work are not clear. There are some phrases like “inspired by” Sinsha et, al. 2022, Grover et, al. 2019), or Hu et, al. 2021. but it is not clear from the text what the main contribution is or what the challenge is of applying these techniques to MARL.
>
> We appreciate the opportunity to clarify this point. While our method is indeed inspired by density-ration estimation approaches such as Grover et al. (2019) and Sinha et al. (2022), applying such techniques to MARL introduces several challenges that have not been addressed in prior work. 1) No tractable joint policy distribution available in MARL. In single-agent RL, density ratios are computed directly using policy probabilities over actions. In MARL, however, the joint action and joint observation space grows exponentially with the number of agents, making it infeasible to compute joint policy densities $ \pi(a|s) $ explicitly. 2) Replay buffer data is generated by a continuously changing joint policy rather than a single stationary behavior policy. This introduces severe distribution mixing in the buffer that standard density-ratio estimation methods are not designed to handle. 3) Need for state-action conditioned $\lambda$ at per-step granularity. Previous TD($\lambda$) works typically consider fixed $\lambda$ or schedule-driven $\lambda$. In contrast, we produce a sample-level adaptive $\lambda$​ based on the relative likelihood of each transition under the current policy, which, to our knowledge, has not been applied before in MARL.
>
> Thus, although conceptually related to prior density-ratio estimation work, the contributions of this paper include 1) A MARL-specific formulation of adaptive $\lambda$ based on likelihood-free off-policy estimation, usable with both value-based and actor-critic MARL. 2) A training mechanism using two replay buffers to approximate on- and off-policy distributions without explicit policy modeling. 3) Demonstration that this method improves convergence speed and performance across multiple SMAC and GRF scenarios. We updated that in the revised version in orange color.
>
> > W2: Related to the first point, if the off-policyness can be computed over states and joint actions, then it is a simple application of previous techniques to MARL. There is nothing specific to MARL which makes it hard to apply these techniques, such as factorized policies or the exponential complexity of the joint state/action space.
>
> Based on the response to Weakness 1, we respectfully disagree and clarify that Direct computation of joint policy probabilities is computationally infeasible in MARL, especially for factored or decentralized policies. Existing off-policy estimation methods assume that $\omega(s,a) = \frac{\pi(a|s)}{\mu(a|s)}$ but neither $\pi$ nor $\mu$ is available in closed form in decentralized MARL. Our approach achieves this by learning $\omega(s,a)$​ from replay data alone, which is a key novelty compared to prior approaches. Additionally, value decomposition methods, such as QMIX, require that bootstrapped TD targets do not violate monotonicity. Plugging learned $\lambda$ values into these architectures while preserving contraction assumptions is also important.

---

> ### Author Response · Authors · 2025-11-27
> **Respond to Weaknesses (2/2)**
>
> > W3: The experiments are limited to saturated benchmarks such as SMAC and GRF. Although the authors acknowledge that it cannot be applied well to SMACv2, that is a significant limitation as it is recently a more standard benchmark than SMACv1. The reason for this limitation is not provided in Appendix C (as they mention in the Conclusion). I suggest the authors go deeper into why this limitation persists, as it might provide hints for some challenges in MARL, and later some additional modifications to the current approaches which will be more novel in the MARL context.
>
> Sorry for the confusion and our carelessness. The results and analysis of SMACv2 are missing in previous versions. We placed the contents of GRF in Appendix C and forgot to show the contents of SMACv2. In the current revised version uploaded, the results and the analysis are shown in Appendix D. We would like to summarize the content of the SMACv2 section here.
>
> | ENV           | QMIX  | QPLEX     | WQMIX | MAPPO     | MACPF | ATD+QMIX  |
> | ------------- | ----- | --------- | ----- | --------- | ----- | --------- |
> | terran_5v5    | 53.41 | 56.16     | 41.01 | 42.28     | 25.18 | **57.32** |
> | zerg_5v5      | 27.14 | 30.96     | 19.75 | **50.39** | 16.05 | 32.06     |
> | protoss_5v5   | 53.93 | 59.34     | 35.95 | 50.85     | 22.47 | **60.77** |
> | protoss_5v6   | 6.51  | 7.88      | 3.23  | 6.93      | 0.59  | **8.33**  |
> | protoss_10v10 | 58.68 | 64.06     | 42.81 | 50.69     | 5.25  | **67.47** |
> | protoss_10v11 | 27.37 | **35.99** | 14.25 | 16.28     | 0.65  | 33.34     |
>
> We evaluate our ATD methods on six SMACv2 configurations with default settings and compute performance across 10 million time steps averaged over five seeds. Results show that our method brings small improvements, slightly faster convergence and higher performance in several scenarios. However, our method generally performs similarly to baseline algorithms and does not significantly improve or harm performance overall. The performance overview demonstrates that the adaptive $\lambda$ method competes well with value-based baselines, though in some cases, such as Zerg 5v5, other algorithms like MAPPO perform better. In the discussion, we claim that SMACv2 introduces substantial randomness in unit types, counts, and positions. This high variance reduces repeated transitions between the slow and fast buffer, causes $\lambda$ values to decay quickly, and leads to updates similar to standard TD learning. As a result, in SMACv2, our ATD method performs at least **no worse** than baseline algorithms.
>
> > W4: In the background section, the “true state of the environment” in line 132 is not related to centralized training. The Dec-POMDP is a formulation, and centralized training is a choice on how to train policies. Similar comment holds for line 137 where partial observability is a characteristic of the formulation/setup of the environment, and it is not related to decentralized execution. Please separate these in the text as it will confuse readers.
> >
> > W5: The observations for $z$ line 136 should have agent Id conditioning.
>
> We fully agree with the reviewer, and thanks for this suggestion. We have clarified that in the Background section in this revised version in orange color.
>
> > W6: The notation $T$ is used twice to indicate the observation-action history and the target value in Eq.1.
>
> We agree and will update symbols to avoid overloading $T$ as both the trajectory and target value. We have changed the observation-action history from $T$ to $H$, which has been corrected in the revised version.

---

> ### Author Response · Authors · 2025-11-27
> **Respond to Questions**
>
> > Q1: What is the connection between off-policyness and $\lambda$? What is the justification behind Eq. 9-10? My understanding is that if the density ratio $\omega$ is high, we have higher off-policyness and thus we would want to go more towards TD(0), but it seems like the reverse is happening in Eq. 9-10.
>
> Sorry for the confusion. First, I want to clarify that the off-policyness $\omega$ is used as $\lambda$ value in Eq.9 and Eq.10, and we have corrected the notation in the equation in this revised version.
>
> The density ratio $\omega$ is defined as $\omega(s,a) = \frac{d^\pi(s,a)}{d^D(s,a)}$. Therefore, a higher $\omega$ value means the sample is similar to the current policy (more on-policy), and in contrast lower $\omega$ value means that the $d^\pi$ is 'different from' $d^D$ and the sampled transition comes from an older behavior policy. In such a way, lower $\omega$ in Eq.10 means older trajectory -> higher TD accuracy and lower MC return accuracy -> smaller $R_{t+1}^\omega$ (MC) weight and larger $\max Q_{tot}$ (TD) weight. Eq.9 is the equation to calculate TD error to update the utility network.
>
>
>
> We thank the reviewer again for the helpful feedback and will incorporate the improvements into the final manuscript. We believe that addressing these points will provide a clearer understanding of our work and its contribution to the field. We are more than willing to provide any further information or clarification that might be required for a more comprehensive evaluation.

---

### Official Review · Reviewer_7bbf · 2025-10-31

**Soundness:** 3
**Presentation:** 2
**Contribution:** 3
**Rating:** 4
**Confidence:** 4

**Summary:**

This paper proposes a method called ATD(λ) for adaptively determining the λ value in TD(λ) for multi-agent reinforcement learning (MARL). In traditional methods, λ is a hyperparameter that needs to be preset, whereas this paper's method dynamically assigns appropriate λ values for each transition by calculating the likelihood of state-action pairs under the current policy. The authors use a likelihood-free density ratio estimator and two replay buffers of different sizes (one for off-policy data and one for approximating on-policy data) to estimate the "on-policy degree" of sampled transitions. The method was evaluated on SMAC (StarCraft Multi-Agent Challenge) and Google Football tasks, showing improved performance compared to fixed λ values.

**Strengths:**

1. The paper addresses an important hyperparameter tuning issue in MARL by making the λ value adaptive rather than fixed, reducing the burden of manual parameter tuning.

2. The proposed method applies to different types of MARL algorithms, including value-based algorithms (like QMIX) and actor-critic-based algorithms (like MAPPO), demonstrating good adaptability.

3. The authors provide a detailed theoretical analysis (Section 4.2 and Appendix A), connecting their method with f-divergence and importance sampling, providing solid theoretical foundations for the proposed algorithm.

**Weaknesses:**

1. In Figure 3, the authors show winning rate curves but do not sufficiently explain why adaptive λ values lead to performance improvements. There's a lack of intuitive explanation of how λ values affect learning dynamics.

2. The text in the picture is too small; even when enlarged, it’s still unreadable. In Figure 4, the curves are difficult to distinguish, especially in the super-hard task scenarios, making it challenging to interpret performance comparisons.

3. Figure 6 shows that the ratio between on-policy and off-policy buffer sizes is a critical hyperparameter with a significant impact. The authors chose 50x as the default ratio based on empirical observation, but did not provide theoretical guidance or sensitivity analysis to support this choice.

4. The results in Figure 10b show inconsistent effects of applying importance weights across different scenarios - helpful in 10m_vs_11m but detrimental in 6h_vs_8z. This suggests the method may not be robust, but the authors don't provide clear guidance on when to apply importance weights.

**Questions:**

1. How does the method scale to environments with more agents? The largest test case appears to be 27 agents vs. 30, but how would it perform with hundreds of agents?

2. How much does the network architecture of the λ predictor influence performance? Would simpler or more complex networks affect performance?

3. Can this adaptive λ method extend beyond cooperative MARL to other RL paradigms, such as competitive or mixed cooperative-competitive settings?

---

> ### Author Response · Authors · 2025-11-27
> **Respond to Weaknesses**
>
> We thank the reviewer for the constructive comments and valuable suggestions. We now try our best to respond to each point in detail below.
>
> > W1: In Figure 3, the authors show winning rate curves but do not sufficiently explain why adaptive λ values lead to performance improvements. There's a lack of intuitive explanation of how λ values affect learning dynamics.
>
> Thank you for raising this concern. We agree that a clearer intuitive explanation will help readers to understand the role of $\lambda$. The empirical results from Figure 3 demonstrate the overall performance enhancement of our ATD method. In the TD($\lambda$) method,  controls the trade-off between TD updates and MC updates. During the MARL training process, the policy and value estimates shift substantially, creating phases where different forms of return estimation are preferable. For example, during the early training process, the utility network (critic network) is inaccurate. Higher $\lambda$ values (almost MC) allow the target to better reflect real long-term returns, which accelerates the fitting and speeds up the stabilization of the joint value. Then, from the middle to late training process, the policy stabilizes, and replay samples from older policies become biased relative to the current policy. In such a way, smaller $\lambda$ values reduce reliance on outdated MC returns and instead trust the increasingly accurate critic. We also had a discussion in section B of the appendix.
>
> Our ATD method directly measures how close a sampled trajectory is to the current policy distribution. Therefore, more on-policy samples will get a high $\lambda$ value and off-policy samples automatically receive smaller $\lambda$​ values. Thus, the method shifts the learning target continuously in accordance with the actual data distribution, rather than fixing a single $\lambda$ that can be suboptimal for part of training. Experiments in Figures 3–5 empirically demonstrate this effect through faster convergence and higher final performance. We have added this intuitive explanation in the introduction section of the revised version colored in red.
>
> > W2: The text in the picture is too small; even when enlarged, it’s still unreadable. In Figure 4, the curves are difficult to distinguish, especially in the super-hard task scenarios, making it challenging to interpret performance comparisons.
>
> We appreciate this feedback. We find that the cyan lines ($\lambda=0$) are sometimes mixed with green (Direct Importance Sampling) and blue (Retrace) color lines. In this revised version, we change the cyan lines into purple lines in Figure 4, which should enlarge the readability.
>
> > W3: Figure 6 shows that the ratio between on-policy and off-policy buffer sizes is a critical hyperparameter with a significant impact. The authors chose 50x as the default ratio based on empirical observation, but did not provide theoretical guidance or sensitivity analysis to support this choice.
>
> We acknowledge that 50x was selected based on empirical analysis. The main principle is that the small buffer should reflect the most recent policy distribution $d^\pi$ and the large buffer approximates the long-term mixed distribution $d^D$. If the small buffer is too small, it fails to represent the current policy and $\lambda$ becomes noisy, and if it is too large, old data contaminates the on-policy distribution, lowering the meaningful separation between the buffers. While a complete theoretical derivation is non-trivial, we conducted this ablation study showing that performance varies smoothly around 25–100×, and the method is not fragile to small deviations.
>
> > W4: The results in Figure 10b show inconsistent effects of applying importance weights across different scenarios - helpful in 10m_vs_11m but detrimental in 6h_vs_8z. This suggests the method may not be robust, but the authors don't provide clear guidance on when to apply importance weights.
>
> This is a good observation. The main reason for this variation is that the benefit of explicit importance weighting depends on the mismatch between replayed data and the current policy. For example, in 10m\_vs\_11m, policies evolve quickly, and replay data becomes stale rapidly. Importance weighting reduces bias and improves value estimation. In 6h\_vs\_8z, policies shift slowly and transitions remain closer to on-policy; explicit weighting introduces additional variance without bringing benefit. Our $\lambda$ predictor already implicitly captures the usable degree of off-polici­ness. Therefore, explicit importance weighting is optional. As discussed in the appendix, the weighted priority on the training data may cause the training process to focus on certain data, which might harm the exploration process and result in unacceptable results. We show that in red in Appendix E.

---

> ### Author Response · Authors · 2025-11-27
> **Respond to Questions**
>
> > Q1: How does the method scale to environments with more agents? The largest test case appears to be 27 agents vs. 30, but how would it perform with hundreds of agents?
>
> The largest scenario in SMAC is 27\_vs\_30, but the method itself should have the scalability ability. The $\lambda$ estimation is per state-joint action and is computed by a lightweight discriminator-style network. The $\lambda$ calculated by ATD should be more acceptable compared with the direct importance sampling method. Meanwhile, the replay buffers also scale linearly with the number of stored transitions. As far as I am concerned, based on the CTDE setting, the primary bottleneck in MARL scaling remains the joint value function and mixer architecture, not our $\lambda$ module. Since the $\lambda$ predictor processes batches on GPU, it scales well to more agents provided the underlying MARL framework can handle the expanded joint observation/state.
>
> > Q2: How much does the network architecture of the $\lambda$ predictor influence performance? Would simpler or more complex networks affect performance?
>
> Thank you for the question. The ATD network is responsible for calculating the f-divergence between two density ratios based on the state values and the transition history within a trajectory.  In such a way, we believe that a network for encoding the history is needed, and the GRU cell is needed. Meanwhile, the feed-forward network reflects history encoding to the $\lambda$ value nonlinearly. Larger network parameters enable larger encoding and reflection abilities, which should improve the performance slightly. However, a larger network size also requires larger training data. The most benefits come from separating replay data statistically and computing the relative state-action likelihood, not from the design of a sophisticated network and according to the empirical results, the simpler architectures are viable. We add an ablation section of RNN module in the discussion section.
>
> > Q3: Can this adaptive $\lambda$ method extend beyond cooperative MARL to other RL paradigms, such as competitive or mixed cooperative-competitive settings?
>
> Yes. Our ATD method is not restricted to cooperative settings. The idea of estimating on-policy likelihood via replay distribution and adapting the TD target accordingly should apply whenever off-policy TD methods are used. The adaptive TD($\lambda$) method aims at increasing the value estimation accuracy of the observation-action pairs of agents based on historical data. Therefore, this method should also work in competitive or mixed settings. As for single-agent RL, this method should also work by calculating the density ratio divergence parametrically. Despite that, our method takes more effort in multi-agent scenarios where the importance sampling is hard to calculate via joint policies, which is also the premise of the main problem addressed by this paper.
>
>
>
> We thank the reviewer again for the helpful feedback and will incorporate the improvements into the final manuscript. We believe that addressing these points will provide a clearer understanding of our work and its contribution to the field. We are more than willing to provide any further information or clarification that might be required for a more comprehensive evaluation.

---

### Author Response · Authors · 2025-12-04
**Response to Area Chair**

Dear Area Chair,

Thank you very much for your time and effort in handling our submission. In addition to the reviewer-specific replies and common responses provided below, we briefly summarize the main contribution of our work and how we addressed the reviewers’ concerns, so that the revisions are easier to follow from your perspective as Area Chair.

------

**Main contribution:**

The proposed ATD($\lambda$) framework addresses a fundamental limitation of existing MARL methods that rely on fixed TD($\lambda$) values, which fail to capture the dynamic and non-stationary nature of multi-agent training. In complex environments with large joint action spaces and limited transition data, estimating policy distributions, and thus determining suitable $\lambda$ values via direct importance sampling, becomes infeasible using standard statistical methods. ATD($\lambda$) overcomes this challenge by introducing a likelihood-free, parametric density-ratio estimator that infers the relative on-policiness of each transition using two replay buffers of different horizons. This enables the algorithm to assign adaptive $\lambda$ values to individual state–action pairs, allowing the TD target to adjust continuously to the evolving policy distribution. Our approach integrates seamlessly into both value-based and actor–critic MARL frameworks, requiring only minor implementation modifications to existing training pipelines. Through extensive experiments on SMAC and Google Football academy scenarios, we demonstrate that equipping QMIX and MAPPO with ATD($\lambda$) consistently yields superior or highly competitive performance compared to their fixed-$\lambda$ counterparts, highlighting the practical effectiveness and general applicability of our method.

------

**Summary of rebuttal:** It is unfortunate that we could not receive follow-up comments from most reviewers due to an OpenReview system issue. We sincerely thank the reviewers for their thoughtful comments and constructive feedback. Across all reviews, concerns primarily focused on the clarity of contributions, the novelty of applying ATD in multi-agent reinforcement learning (MARL), the empirical evaluation, and technical explanations regarding the connection between adaptive $\lambda$ and off-policy correction. We clarify that, while ATD is conceptually inspired by prior density-ratio and off-policy correction methods, its adaptation to MARL introduces significant challenges not addressed in single-agent settings. Specifically, the exponential growth of joint state-action spaces, non-stationarity of joint policies, and difficulty in computing tractable joint distributions make standard density-ratio estimation infeasible. Our contributions include a MARL-specific formulation of adaptive $\lambda$ via a likelihood-free off-policy estimator, a two-buffer training mechanism for sample-level trace adaptation, and empirical validation showing improved convergence and competitive performance across SMAC, SMACv2, and GFootball (GRF) benchmarks. These innovations enable stable and efficient integration into both value-based (QMIX) and actor-critic (MAPPO) algorithms, demonstrating practical utility in multi-agent domains.

Regarding technical and empirical concerns, we clarified the relationship between the learned $\lambda$ and off-policyness: lower density-ratio values indicate older or off-policy transitions, which automatically increase TD weighting and reduce Monte Carlo contribution, yielding a bias-variance tradeoff aligned with classical TD theory. Meanwhile, the empirical evidence confirms that our learned $\lambda$ improves convergence speed, and consistently competes with or outperforms strong baselines. We also addressed concerns about the integration with QMIX and MAPPO, the use of importance sampling as a diagnostic measure, the explanation of Jensen-Shannon divergence in density estimation, and presentation issues including figure captions and algorithm clarity. Finally, limitations such as partial performance in SMACv2 due to stochasticity are explicitly analyzed, showing that ATD remains at least as effective as baseline methods. Collectively, these clarifications highlight the novelty, soundness, and practical contributions of our work in advancing adaptive TD learning for multi-agent reinforcement learning.

In summary, we believe that ATD offers clear novelty and a distinct contribution. During the rebuttal period, the central theoretical concerns were substantially resolved, and the extended experiments and ablations have further strengthened the paper and addressed the main points raised by the reviewers. The changes are shown in the latest revised version. We hope this overview is helpful for your decision, and we are sincerely grateful for your careful consideration of our work.

---

### Note · Authors · 2026-01-20

I have read and agree with the venue's withdrawal policy on behalf of myself and my co-authors.